# Energy-Based Models for Anomaly Detection:
# A Manifold Diffusion Recovery Approach

**Sangwoong Yoon**
Korea Institute for Advanced Study
swyoon@kias.re.kr

**Young-Uk Jin**
Samsung Electronics
yueric.jin@samsung.com

**Yung-Kyun Noh**[*]
Hanyang University
Korea Institute for Advanced Study
nohyung@hanyang.ac.kr

**Frank C. Park**[*]
Seoul National University
Saige Research
fcp@snu.ac.kr

## Abstract

We present a new method of training energy-based models (EBMs) for anomaly detection that leverages low-dimensional structures within data. The proposed algorithm, Manifold Projection-Diffusion Recovery (MPDR), first perturbs a data point along a low-dimensional manifold that approximates the training dataset. Then, EBM is trained to maximize the probability of recovering the original data. The training involves the generation of negative samples via MCMC, as in conventional EBM training, but from a different distribution concentrated near the manifold. The resulting near-manifold negative samples are highly informative, reflecting relevant modes of variation in data. An energy function of MPDR effectively learns accurate boundaries of the training data distribution and excels at detecting out-of-distribution samples. Experimental results show that MPDR exhibits strong performance across various anomaly detection tasks involving diverse data types, such as images, vectors, and acoustic signals.

## 1 Introduction

Unsupervised detection of anomalous data appears frequently in practical applications, such as industrial surface inspection [1], machine fault detection [2, 3], and particle physics [4]. Modeling the probability distribution of normal data $p_{data}(\mathbf{x})$ is a principled approach for anomaly detection [5]. Anomalies, often called outliers or out-of-distribution (OOD) samples, lie outside of the data distribution and can thus be characterized by low probability density under the distribution. However, many deep generative models that can evaluate the likelihood of data, including variational autoencoders (VAE; [6]), autoregressive models [7], and flow-based models [8] are known to perform poorly on popular anomaly detection benchmarks such as CIFAR-10 (in) vs SVHN (out), by assigning high likelihood on seemingly trivial outliers [9, 10].

On the other hand, deep energy-based models (EBMs) have demonstrated notable improvement in anomaly detection compared to other deep generative models [11]. While the specific reason for the superior performance of EBMs has not been formally analyzed, one probable factor is the explicit mechanism employed in EBM's maximum likelihood training that reduces the likelihood of negative samples. These negative samples are generated from the model distribution $p_\theta(\mathbf{x})$ using Markov Chain Monte Carlo (MCMC). Since modern EBMs operate in high-dimensional data spaces, it is extremely difficult to cover the entire space with a finite-length Markov chain. The difficulty in

---

[*]Corresponding authors

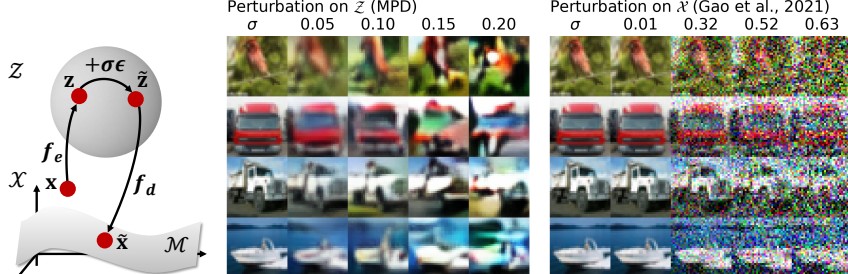

Figure 1: (Left) An illustration of Manifold Projection-Diffusion. A datum $\mathbf{x}$ is first projected into the latent space $\mathcal{Z}$ through the encoder $f_e(\mathbf{x})$ and then diffused with a local noise, such as Gaussian. The perturbed sample $\tilde{\mathbf{x}}$ is obtained by projecting it back to the input space $\mathcal{X}$ through the decoder $f_d(\mathbf{z})$. (Right) Comparison between the perturbations in MPDR and DRL [16] on CIFAR-10 examples.

generating negative samples triggered the development of several heuristics such as using truncated chains [12], persistent chains [13], sample replay buffers [11], and data augmentation in the middle of the chain [14].

Instead of requiring a Markov chain to cover the entire space, EBM can be trained with MCMC running within the vicinity of each training datum. For example, Contrastive Divergence (CD; [15]) uses a short Markov chain initialized on training data to generate negative samples close to the training distribution. Diffusion Recovery Likelihood (DRL; [16]) enables MCMC to focus on the training data's neighborhood using a different objective function called recovery likelihood. Recovery likelihood $p_\theta(\mathbf{x}|\tilde{\mathbf{x}})$ is the conditional probability of training datum $\mathbf{x}$ given the observation of its copy $\tilde{\mathbf{x}}$ perturbed with a Gaussian noise. Training of DRL requires sampling from the recovery likelihood distribution $p_\theta(\mathbf{x}|\tilde{\mathbf{x}})$, which is easier than sampling from the model distribution $p_\theta(\mathbf{x})$, as $p_\theta(\mathbf{x}|\tilde{\mathbf{x}})$ is close to uni-modal and concentrated near $\mathbf{x}$. While CD and DRL significantly stabilize the negative sampling process, the resulting EBMs exhibit limited anomaly detection performance due to the insufficient coverage of negative samples in the input space.

In this paper, we present a novel training algorithm for EBM that does not require MCMC covering the entire space while achieving strong anomaly detection performance. The proposed algorithm, **Manifold Projection-Diffusion Recovery** (MPDR), extends the recovery likelihood framework by replacing Gaussian noise with **Manifold Projection-Diffusion** (MPD), a novel perturbation operation that reflects low-dimensional structure of data. Leveraging a separately trained autoencoder manifold, MPD firstly projects a training datum onto the manifold and perturbs the datum along the manifold. Compared to Gaussian noise, MPD reflects relevant modes of variation in data, such as change in colors or shapes in an image, as shown in Fig. 1. A MPD-perturbed sample serves as an informative starting point for MCMC that generates a negative sample, teaching EBM to discriminate challenging outliers that has partial similarity to training data.

Given an MPD-perturbed sample, EBM is trained by maximizing the recovery likelihood. We derive a simple expression for evaluating the recovery likelihood for MPD. We also propose an efficient two-stage MCMC strategy for sampling from the recovery likelihood, leveraging the latent space of the autoencoder. Additionally, we show that the model parameter estimation by MPDR is consistent under standard assumptions.

Compared to existing EBM training algorithms using an autoencoder-like auxiliary module [17, 18, 19, 20], MPDR has two advantages. First, MPDR allows the use of multiple autoencoders for training. The manifold ensemble technique stabilizes the training and improves anomaly detection performance by providing diverse negative samples. Second, MPDR achieves good anomaly detection performance with lightweight autoencoders. For example, autoencoders in our CIFAR-10 experiment has about only 1.5~3 million parameters, which are significantly fewer than that of NVAE [21] (~130 million) or Glow [8] (~44 million), both of which are previously used in EBM training [17, 18].

Our contributions can be summarized as follows:

- We propose MPDR, a novel method of using autoencoders for training EBM. MPDR is compatible with any energy functions and gives consistent density parameter estimation with any autoencoders.
- We provide a suite of practical strategies for achieving successful anomaly detection with MPDR, including two-stage sampling, energy function design, and ensembling multiple autoencoders.

- We demonstrate the effectiveness of MPDR on unsupervised anomaly detection tasks with various data types, including images, representation vectors, and acoustic signals.

## 2   Preliminaries

**Energy-Based Models (EBM)**   An energy-based generative model, or an unnormalized probabilistic model, represents a probability density function using a scalar energy function $E_\theta : \mathcal{X} \to \mathbb{R}$, where $\mathcal{X}$ denotes the domain of data. The energy function $E_\theta$ defines a probability density function $p_\theta$ through the following relationship:

$$p_\theta(\mathbf{x}) \propto \exp(-E_\theta(\mathbf{x})). \tag{1}$$

The parameters $\theta$ can be learned by maximum likelihood estimation given iid samples from the underlying data distribution $p_{data}(\mathbf{x})$. The gradient of the log-likelihood for a training sample $\mathbf{x}$ is well-known [15] and can be written as follows:

$$\nabla_\theta \log p_\theta(\mathbf{x}) = -\nabla_\theta E_\theta(\mathbf{x}) + \mathbb{E}_{\mathbf{x}^- \sim p_\theta(\mathbf{x})}[\nabla_\theta E_\theta(\mathbf{x}^-)], \tag{2}$$

where $\mathbf{x}^-$ denotes a "negative" sample drawn from the model distribution $p_\theta(\mathbf{x})$. Typically, $\mathbf{x}^-$ is generated using Langevin Monte Carlo (LMC). In LMC, points are arbitrarily initialized and then iteratively updated in a stochastic manner to simulate independent sampling from $p_\theta(\mathbf{x})$. For each time step $t$, a point $\mathbf{x}^{(t)}$ is updated by $\mathbf{x}^{(t+1)} = \mathbf{x}^{(t)} - \lambda_1 \nabla_{\mathbf{x}} E_\theta(\mathbf{x}^{(t)}) + \lambda_2 \epsilon^{(t)}$, for $\epsilon^{(t)} \sim \mathcal{N}(0, \mathbf{I})$. The step size $\lambda_1$ and the noise scale $\lambda_2$ are often tuned separately in practice. Since LMC needs to be performed in every iteration of training, it is infeasible to run negative sampling until convergence, and a compromise must be made. Popular heuristics include initializing MCMC on training data [15], using short-run LMC [12], and utilizing a replay buffer [13, 11].

**Recovery Likelihood [22, 16]**   Instead of directly maximizing the likelihood (Eq. (2)), $\theta$ can be learned through the process of denoising data from artificially injected Gaussian noises. Denoising corresponds to maximizing the *recovery likelihood* $p(\mathbf{x}|\tilde{\mathbf{x}})$, the probability of recovering data $\mathbf{x}$ from its noise-corrupted version $\tilde{\mathbf{x}} = \mathbf{x} + \sigma\epsilon$, where $\epsilon \sim \mathcal{N}(0, \mathbf{I})$.

$$p_\theta(\mathbf{x}|\tilde{\mathbf{x}}) \propto p_\theta(\mathbf{x})p(\tilde{\mathbf{x}}|\mathbf{x}) \propto \exp\left(-E_\theta(\mathbf{x}) - \frac{1}{2\sigma^2}||\mathbf{x} - \tilde{\mathbf{x}}||^2\right), \tag{3}$$

where Bayes' rule is applied. The model parameter $\theta$ is estimated by maximizing the log recovery likelihood, i.e., $\max_\theta \mathbb{E}_{\mathbf{x},\tilde{\mathbf{x}}}[\log p_\theta(\mathbf{x}|\tilde{\mathbf{x}})]$, for $\mathbf{x} \sim p_{data}(\mathbf{x}), \tilde{\mathbf{x}} \sim p(\tilde{\mathbf{x}}|\mathbf{x})$, where $p(\tilde{\mathbf{x}}|\mathbf{x}) = \mathcal{N}(\mathbf{x}, \sigma^2\mathbf{I})$. This estimation is shown to be consistent under the usual assumptions (Appendix A.2 in [16]). DRL [16] uses a slightly modified perturbation $\tilde{\mathbf{x}} = \sqrt{1 - \sigma^2}\mathbf{x} + \sigma\epsilon$ in training EBM, following [23]. This change introduces a minor modification of Eq. (3).

The recovery likelihood $p_\theta(\mathbf{x}|\tilde{\mathbf{x}})$ (Eq. 3) is essentially a new EBM with the energy $\tilde{E}_\theta(\mathbf{x}|\tilde{\mathbf{x}}) = E_\theta(\mathbf{x}) + ||\mathbf{x} - \tilde{\mathbf{x}}||^2/2\sigma^2$. Therefore, the gradient $\nabla_\theta \log p_\theta(\mathbf{x}|\tilde{\mathbf{x}})$ has the same form with the log-likelihood gradient of EBM (Eq. (2)), except that negative samples are drawn from $p_\theta(\mathbf{x}|\tilde{\mathbf{x}})$ instead of $p_\theta(\mathbf{x})$:

$$\nabla_\theta \log p_\theta(\mathbf{x}|\tilde{\mathbf{x}}) = -\nabla_\theta E_\theta(\mathbf{x}) + \mathbb{E}_{\mathbf{x}^- \sim p_\theta(\mathbf{x}|\tilde{\mathbf{x}})}[\nabla_\theta E_\theta(\mathbf{x}^-)], \tag{4}$$

where $\nabla_\theta \tilde{E}_\theta(\mathbf{x}|\tilde{\mathbf{x}}) = \nabla_\theta E_\theta(\mathbf{x})$ as the Gaussian noise is independent of $\theta$. Sampling from $p_\theta(\mathbf{x}|\tilde{\mathbf{x}})$ is more stable than sampling from $p_\theta(\mathbf{x})$, because $p_\theta(\mathbf{x}|\tilde{\mathbf{x}})$ is close to a uni-modal distribution concentrated near $\mathbf{x}$ [16, 22]. However, it is questionable whether Gaussian noise is the most informative way to perturb data in a high-dimensional space.

## 3   Manifold Projection-Diffusion Recovery

We introduce the Manifold Projection-Diffusion Recovery (MPDR) algorithm, which trains EBM by recovering from perturbations that are more informative than Gaussian noise. Firstly, we propose Manifold Projection-Diffusion (MPD), a novel perturbation operation leveraging the low-dimensional structure inherent in the data. Then, we derive the recovery likelihood for MPD. We also provide an efficient sampling strategy and practical implementation techniques for MPDR. The implementation of MPDR is publicly available[1].

---

[1] https://github.com/swyoon/manifold-projection-diffusion-recovery-pytorch

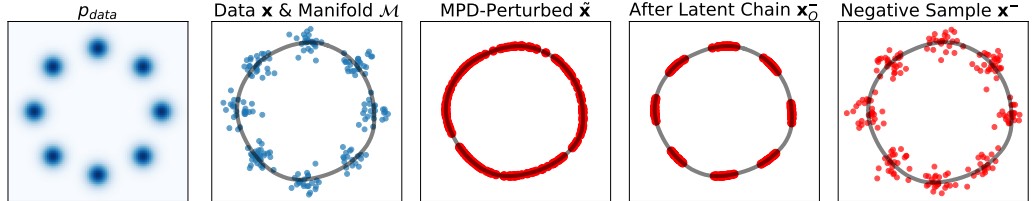

Figure 2: Negative sample generation process in MDPR. Data $\mathbf{x}$ (blue dots) are first projected and diffused on autoencoder manifold $\mathcal{M}$ (gray line), resulting in $\tilde{\mathbf{x}}$. The latent chain starting from $\tilde{\mathbf{x}}$ generates $\mathbf{x}_0^- \in \mathcal{M}$. The visible chain draws negative samples $\mathbf{x}^-$ from $p(\mathbf{x}|\tilde{\mathbf{x}})$ starting from $\mathbf{x}_0^-$.

**Autoencoders**   MPDR assumes that a pretrained autoencoder approximating the training data manifold is given. The autoencoder consists of an encoder $f_e : \mathcal{X} \to \mathcal{Z}$ and a decoder $f_d : \mathcal{Z} \to \mathcal{X}$, both assumed to be deterministic and differentiable. The latent space is denoted as $\mathcal{Z}$. The dimensionalities of $\mathcal{X}$ and $\mathcal{Z}$ are denoted as $D_{\mathbf{x}}$ and $D_{\mathbf{z}}$, respectively. We assume $f_e$ and $f_d$ as typical deep neural networks jointly trained to reduce the training data's reconstruction error $l(\mathbf{x})$, where $l(\mathbf{x})$ is typically an $l_2$ error, $l(\mathbf{x}) = ||\mathbf{x} - f_d(f_e(\mathbf{x}))||^2$.

## 3.1   Manifold Projection-Diffusion

We propose a novel perturbation operation, **Manifold Projection-Diffusion** (MPD). Instead of adding Gaussian noise directly to a datum $\mathbf{x} \xrightarrow{+\sigma\epsilon} \tilde{\mathbf{x}}$ as in the conventional recovery likelihood, MPD first encodes $\mathbf{x}$ using the autoencoder and then applies a noise in the latent space:

$$\mathbf{x} \xrightarrow{f_e} \mathbf{z} \xrightarrow{+\sigma\epsilon} \tilde{\mathbf{z}} \xrightarrow{f_d} \tilde{\mathbf{x}}, \tag{5}$$

where $\mathbf{z} = f_e(\mathbf{x})$, $\tilde{\mathbf{z}} = \mathbf{z} + \sigma\epsilon$, and $\tilde{\mathbf{x}} = f_d(\tilde{\mathbf{z}})$. The noise magnitude $\sigma$ is a predefined constant and $\epsilon \sim \mathcal{N}(0, \mathbf{I})$. The first step **projects** $\mathbf{x}$ into $\mathcal{Z}$, and the second step **diffuses** the encoded vector $\mathbf{z}$. When decoded through $f_d$, the output $\tilde{\mathbf{x}}$ always lies on the **decoder manifold** $\mathcal{M} = \{\mathbf{x}|\mathbf{x} = f_d(\mathbf{z}), \mathbf{z} \in \mathcal{Z}\}$, a collection of all possible outputs from the decoder $f_d(\mathbf{z})$. The process is visualized in the left panel of Fig. 1.

Since $\mathbf{z}$ serves as a coordinate of $\mathcal{M}$, a Gaussian noise in $\mathcal{Z}$ corresponds to perturbation of data along the manifold $\mathcal{M}$, reflecting more relevant modes of variation in data than Gaussian perturbation in $\mathcal{X}$ (Fig. 1). Note that MPD reduces to the conventional Gaussian perturbation if we set $\mathcal{Z} = \mathcal{X}$ and set $f_e$ and $f_d$ as identity mappings.

## 3.2   Manifold Projection-Diffusion Recovery Likelihood

We define the recovery likelihood for MPD as $p_\theta(\mathbf{x}|\tilde{\mathbf{z}})$, which is evaluated as follows:

$$p_\theta(\mathbf{x}|\tilde{\mathbf{z}}) \overset{(i)}{\propto} p_\theta(\mathbf{x})p(\tilde{\mathbf{z}}|\mathbf{x}) = p_\theta(\mathbf{x})\left(\int p(\tilde{\mathbf{z}}|\mathbf{z})p(\mathbf{z}|\mathbf{x})\mathrm{d}\mathbf{z}\right) \overset{(ii)}{=} p_\theta(\mathbf{x})\left(\int p(\tilde{\mathbf{z}}|\mathbf{z})\delta_{f_e(\mathbf{x})}(\mathbf{z})\mathrm{d}\mathbf{z}\right) \tag{6}$$

$$\overset{(iii)}{=} p_\theta(\mathbf{x})p(\tilde{\mathbf{z}}|\mathbf{z} = f_e(\mathbf{x})) \propto \exp(-E_\theta(\mathbf{x}) + \log p(\tilde{\mathbf{z}}|\mathbf{z})) \equiv \exp(-\tilde{E}_\theta(\mathbf{x}|\tilde{\mathbf{z}})). \tag{7}$$

In (i), we apply Bayes' rule. In (ii), we treat $p(\mathbf{z}|\mathbf{x})$ as $\delta_{\mathbf{z}}(\cdot)$, a Dirac measure on $\mathcal{Z}$ at $\mathbf{z} = f_e(\mathbf{x})$, as our encoder is deterministic. Equality (iii) results from the property of Dirac measure $\int f(\mathbf{x})\delta_{\mathbf{y}}(\mathbf{z})\mathrm{d}\mathbf{z} = f(\mathbf{y})$. Using Gaussian perturbation simplifies the energy to $\tilde{E}_\theta(\mathbf{x}|\tilde{\mathbf{z}}) = E_\theta(\mathbf{x}) + \frac{1}{2\sigma^2}||\tilde{\mathbf{z}} - f_e(\mathbf{x})||^2$.

Now, $E_\theta(\mathbf{x})$ is trained by maximizing $\log p_\theta(\mathbf{x}|\tilde{\mathbf{z}})$. The gradient of $\log p_\theta(\mathbf{x}|\tilde{\mathbf{z}})$ with respect to $\theta$ has the same form as the conventional maximum likelihood case (Eq. 2) and the Gaussian recovery likelihood case (Eq. 4) but with a different negative sample distribution $p_\theta(\mathbf{x}|\tilde{\mathbf{z}})$:

$$\nabla_\theta \log p_\theta(\mathbf{x}|\tilde{\mathbf{z}}) = -\nabla_\theta E_\theta(\mathbf{x}) + \mathbb{E}_{\mathbf{x}^- \sim p_\theta(\mathbf{x}|\tilde{\mathbf{z}})}[\nabla_\theta E_\theta(\mathbf{x}^-)]. \tag{8}$$

Negative samples $\mathbf{x}^-$ are drawn from $p_\theta(\mathbf{x}|\tilde{\mathbf{z}})$ using MCMC.

Another possible definition for the recovery likelihood is $p(\mathbf{x}|\tilde{\mathbf{x}})$, which becomes equivalent to $p(\mathbf{x}|\tilde{\mathbf{z}})$ when $f_d$ is an injective function, i.e., no two instances of $\tilde{\mathbf{z}}$ map to the same $\tilde{\mathbf{x}}$. However, if $f_d$ is not injective, additional information loss may occur and becomes difficult to compute. As a result, $p(\mathbf{x}|\tilde{\mathbf{z}})$ serves as a more general and also convenient choice for the recovery likelihood.

**Algorithm 1** Manifold Projection-Diffusion Recovery

1: **while** converged **do**
2:     Sample a mini-batch of positive samples $\mathbf{x}$
3:     Compute $\tilde{\mathbf{z}} = f_e(\mathbf{x}) + \sigma\epsilon$
4:     Sample $\mathbf{z}^-$ from energy $\tilde{H}_\theta(\mathbf{z}|\tilde{\mathbf{z}})$ using
       LMC on $\mathcal{Z}$ starting from $\tilde{\mathbf{z}}$
5:     Sample $\mathbf{x}^-$ from energy $\tilde{E}_\theta(\mathbf{x}|\tilde{\mathbf{z}})$ using
       LMC on $\mathcal{X}$ starting from $\mathbf{x}_0^- = f_d(\mathbf{z}^-)$
6:     Update $\theta$ with the gradient:
       $-\frac{\partial}{\partial\theta}E_\theta(\mathbf{x}) + \frac{\partial}{\partial\theta}E_\theta(\mathbf{x}^-)$
7: **end while**

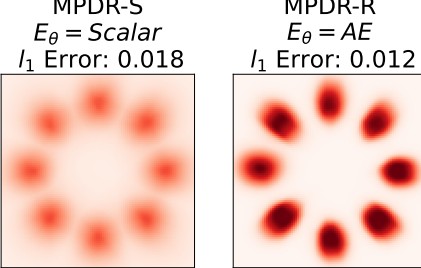

MPDR-S      MPDR-R
$E_\theta = Scalar$    $E_\theta = AE$
$l_1$ Error: 0.018    $l_1$ Error: 0.012

Figure 3: 2D density estimation with a scalar energy function (left) and an autoencoder-based energy function (right).

**Consistency** Maximizing $\log p(\mathbf{x}|\tilde{\mathbf{z}})$ leads to the consistent estimation of $\theta$. The consistency is shown by transforming the objective function into KL divergence. The assumptions resemble those for maximum likelihood estimation, including infinite data, identifiablility, and a correctly specified model. Additionally, the autoencoder ($f_e, f_d$) and the noise magnitude $\sigma$ should be independent of $\theta$, remaining fixed during training. The consistency holds for any choices of ($f_e, f_d$) and $\sigma$, as long as the recovery likelihood $p_\theta(\mathbf{x}|\tilde{\mathbf{z}})$ is non-zero for any $(\mathbf{x}, \tilde{\mathbf{z}})$. Further details can be found in the Appendix.

### 3.3 Two-Stage Sampling

A default method for drawing $\mathbf{x}^-$ from $p_\theta(\mathbf{x}|\tilde{\mathbf{z}})$ is to execute LMC on $\mathcal{X}$, starting from $\tilde{\mathbf{x}}$, as done in DRL [16]. While this **visible chain** should suffice in principle with infinite chain length, sampling can be improved by leveraging the latent space $\mathcal{Z}$, as demonstrated in [24, 18, 20, 25]. For MPDR, we propose a **latent chain**, a short LMC operating on $\mathcal{Z}$ that generates a better starting point $\mathbf{x}_0^-$ for the visible chain. We first define the auxiliary latent energy $\tilde{H}_\theta(\mathbf{z}) = \tilde{E}_\theta(f_d(\mathbf{z})|\tilde{\mathbf{z}})$, the pullback of $\tilde{E}_\theta(\mathbf{x}|\tilde{\mathbf{z}})$ through the decoder $f_d(\mathbf{z})$. Then, we run a latent LMC that samples from a probability density proportional to $\exp(-\tilde{H}_\theta(\mathbf{z}))$. The latent chain's outcome, $\mathbf{z}_0^-$, is fed to the decoder to produce the visible chain's starting point $\mathbf{x}_0^- = f_d(\mathbf{z})$, which is likely to have a smaller energy than the original starting point $\tilde{E}_\theta(\mathbf{x}_0^-|\tilde{\mathbf{z}}) < \tilde{E}(\tilde{\mathbf{x}}|\tilde{\mathbf{z}})$. Introducing a small number of latent chain steps improves anomaly detection performance in our experiments. A similar latent-space LMC method appears in [24] but requires a sample replay buffer not used in MPDR. Fig.2 illustrates the sampling procedure, and Algorithm1 summarizes the training algorithm.

### 3.4 Perturbation Ensemble

Using *multiple* perturbations simultaneously during MPDR training improves performance and stability, while not harming the consistency. Although the consistency of estimation is independent of the specifics of the perturbation, the perturbation design, such as ($f_e, f_d$) and $\sigma$, have a significant impact on performance in practice. Perturbation ensemble alleviates the risk of adhering to a single suboptimal choice of perturbation.

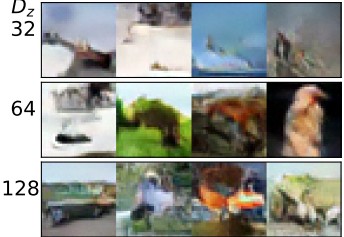

$D_z$
32
64
128

Figure 4: CIFAR-10 negative samples from a manifold ensemble. Negative samples become more visually complex as $D_\mathbf{z}$ grows larger.

**Noise Magnitude Ensemble** We randomly draw the perturbation magnitude $\sigma$ from a uniform distribution over a predefined interval for each sample in a mini-batch. In our implementation, we draw $\sigma$ from the interval $[0.05, 0.3]$ throughout all the experiments.

**Manifold Ensemble** We can also utilize multiple autoencoder manifolds $\mathcal{M}$ in MPD. Given $K$ autoencoders, a minibatch is divided into $K$ equally sized groups. For each group, negative samples are generated separately using the corresponding autoencoder. Only a minimal increase in training time is observed as $K$ increases, since each autoencoder processes a smaller mini-batch. Memory overhead does exist but is manageable, since MPDR can achieve good performance with relatively smaller autoencoders.

Table 1: MNIST hold-out digit detection. Performance is measured in AUPR. The standard deviation of AUPR is computed over the last 10 epochs. The largest mean value is marked in bold, while the second-largest is underlined. Asterisks denote that the results are adopted from literature.

| Hold-Out Digit | 1 | 4 | 5 | 7 | 9 |
|---|---|---|---|---|---|
| Autoencoder (AE) | $0.062 \pm 0.000$ | $0.204 \pm 0.003$ | $0.259 \pm 0.006$ | $0.125 \pm 0.003$ | $0.113 \pm 0.001$ |
| IGEBM | $0.101 \pm 0.020$ | $0.106 \pm 0.019$ | $0.205 \pm 0.108$ | $0.100 \pm 0.042$ | $0.079 \pm 0.015$ |
| MEG [26]* | $0.281 \pm 0.035$ | $0.401 \pm 0.061$ | $0.402 \pm 0.062$ | $0.290 \pm 0.040$ | $0.324 \pm 0.034$ |
| BiGAN-$\sigma$ [27]* | $0.287 \pm 0.023$ | $0.443 \pm 0.029$ | $0.514 \pm 0.029$ | $0.347 \pm 0.017$ | $0.307 \pm 0.028$ |
| Latent EBM[20]* | $0.336 \pm 0.008$ | $0.630 \pm 0.017$ | $0.619 \pm 0.013$ | $0.463 \pm 0.009$ | $0.413 \pm 0.010$ |
| VAE+EBM [19]* | $0.297 \pm 0.033$ | $\underline{0.723} \pm 0.042$ | $0.676 \pm 0.041$ | $0.490 \pm 0.041$ | $0.383 \pm 0.025$ |
| NAE [24] | $\underline{0.802} \pm 0.079$ | $0.648 \pm 0.045$ | $0.716 \pm 0.032$ | $0.789 \pm 0.041$ | $0.441 \pm 0.067$ |
| MPDR-S (ours) | $0.764 \pm 0.045$ | $\mathbf{0.823} \pm 0.018$ | $\underline{0.741} \pm 0.041$ | $\mathbf{0.857} \pm 0.022$ | $\underline{0.478} \pm 0.048$ |
| MPDR-R (ours) | $\mathbf{0.844} \pm 0.030$ | $0.711 \pm 0.029$ | $\mathbf{0.757} \pm 0.024$ | $\underline{0.850} \pm 0.014$ | $\mathbf{0.569} \pm 0.036$ |

An effective strategy is to utilize autoencoders with varying latent space dimensionalities $D_{\mathbf{z}}$. For high-dimensional data, such as images, $\mathcal{M}$ with different $D_{\mathbf{z}}$ tends to capture distinct modes of variation in data. As depicted in Fig. 4, a perturbation on $\mathcal{M}$ with small $D_{\mathbf{z}}$ corresponds to low-frequency variation in $\mathcal{X}$, whereas for $\mathcal{M}$ with large $D_{\mathbf{z}}$, it corresponds to higher-frequency variation. Using multiple $D_{\mathbf{z}}$'s in MPDR gives us more diverse $\mathbf{x}^-$ and eventually better anomaly detection performance.

## 3.5 Energy Function Design

MPDR is a versatile training algorithm for general EBMs, compatible with various types of energy functions. The design of an energy function plays a crucial role in anomaly detection performance, as the inductive bias of an energy governs its behavior in out-of-distribution regions. We primarily explore two designs for energy functions: MPDR-Scalar (**MPDR-S**), a feed-forward neural network that takes input $\mathbf{x}$ and produces a scalar output, and MPDR-Reconstruction (**MPDR-R**), the reconstruction error from an autoencoder, $E_\theta(\mathbf{x}) = ||\mathbf{x} - g_d(g_e(\mathbf{x}))||^2$, for an encoder $g_e$ and a decoder $g_d$. The autoencoder $(g_e, g_d)$ is separate from the autoencoder $(f_e, f_d)$ used in MPD. First proposed in [24], a reconstruction-based energy function has an inductive bias towards assigning high energy values to off-manifold regions (Fig. 3). Training such energy functions using conventional techniques like CD [15] or persistent chains [13, 11] is reported to be challenging [24]. However, MPDR effectively trains both scalar and reconstruction energies. Additionally, in Sec. 4.4, we demonstrate that MPDR is also compatible with an energy function based on a masked autoencoder.

# 4 Experiment

## 4.1 Implementation of MPDR

An autoencoder $(f_e, f_d)$ is trained by minimizing the reconstruction error of the training data and remains fixed during the training of $E_\theta(\mathbf{x})$. When using an ensemble of manifolds, each autoencoder is trained independently. For anomaly detection, the energy value $E_\theta(\mathbf{x})$ serves as an anomaly score which assigns a high value for anomalous $\mathbf{x}$. All optimizations are performed using Adam with a learning rate of 0.0001. Each run is executed on a single Tesla V100 GPU. Other details, including network architectures and LMC hyperparameters, can be found in the Appendix.

Table 2: MNIST OOD detection performance measured in AUPR. We test models from hold-out digit 9 experiment (Table 1). The overall performance is high, as detecting these outliers is easier than identifying the hold-out digit.

| | KMNIST | EMNIST | Omniglot | FashionMNIST | Constant |
|---|---|---|---|---|---|
| AE | 0.999 | 0.977 | 0.947 | 1.000 | 0.954 |
| IGEBM | 0.990 | 0.923 | 0.845 | 0.996 | 1.000 |
| NAE | 1.000 | 0.993 | 0.997 | 1.000 | 1.000 |
| MPDR-S | 0.999 | 0.995 | 0.994 | 0.999 | 1.000 |
| MPDR-R | 0.999 | 0.989 | 0.997 | 0.999 | 0.990 |

**Spherical Latent Space** In all our implementations of autoencoders, we utilize a hyperspherical latent space $\mathcal{Z} = \mathbb{S}^{D_{\mathbf{z}}-1}$ [28, 29, 30]. The encoder output is projected onto $\mathbb{S}^{D_{\mathbf{z}}-1}$ via division by its norm before being fed into the decoder. Employing $\mathbb{S}^{D_{\mathbf{z}}-1}$ standardizes the length scale of

$\mathcal{Z}$, allowing us to use the same value of $\sigma$ across various data sets and autoencoder architectures. Meanwhile, the impact of $\mathbb{S}^{D_\mathbf{z}-1}$ on the reconstruction error is minimal.

**Regularization**  For a scalar energy function, MPDR-S, we add $L_{reg} = E_\theta(\mathbf{x})^2 + E_\theta(\mathbf{x}^-)^2$ to the loss function, as proposed by [11]. For a reconstruction energy function, MPDR-R, we add $L_{reg} = E_\theta(\mathbf{x}^-)^2$, following [24].

**Scaling Perturbation Probability** Applying regularization to an energy restricts its scale, causing a mismatch in scales between the two terms in the recovery likelihood (Eq. (7)). To remedy this mismatch, we heuristically introduce a scale factor $\gamma < 1$ to $\log p(\mathbf{z}|\tilde{\mathbf{z}})$, resulting in the modified recovery energy $\tilde{E}_\theta^{(\gamma)}(\mathbf{x}|\tilde{\mathbf{z}}) = E_\theta(\mathbf{x}) + \frac{\gamma}{2\sigma^2}||\tilde{\mathbf{z}} - f_e(\mathbf{x})||^2$. We use $\gamma = 0.0001$ for all experiments.

## 4.2 2D Density Estimation

We show MPDR's ability to estimate multi-modal density using a mixture of eight circularly arranged 2D Gaussians (Fig. 2). We construct an autoencoder with $\mathcal{S}^1$ latent space, which approximately captures the circular arrangement. The encoder and the decoder are MLPs with two layers of 128 hidden neurons. To show the impact of the design of energy functions, we implement both scalar energy and reconstruction energy. Three-hidden-layer MLPs are used for the scalar energy function, and the encoder and the decoder in the reconstruction energy function. Note that the network architecture of the reconstruction energy is not the same as the autoencoder used for MPD. The density estimation results are presented in Fig. 3. We quantify density estimation performance using $l_1$ error. After numerically normalizing the energy function and true density on the visualized bounded domain, we compute the $l_1$ error at 10,000 grid points. While both energies capture the overall landscape of the density, the reconstruction energy achieves a smaller error by suppressing probability density at off-manifold regions.

## 4.3 Image Out-of-Distribution Detection

**MNIST Hold-Out Digit Detection**  Following the protocol of [26, 27], we evaluated the performance of MPDR on MNIST hold-out digit detection benchmark, where one of the ten digits in the MNIST dataset is considered anomalous, and the remaining digits are treated as in-distribution. This is a challenging task due to the diversity of the in-distribution data and a high degree of similarity between target anomalies and inliers. In particular, selecting digits 1, 4, 5, 7, and 9 as anomalous is known to be especially difficult. The results are shown in Table 1.

In MPDR, we use a single autoencoder $(f_e, f_d)$ with $D_\mathbf{z} = 32$. The energy function of MPDR-S is initialized from scratch, and the energy function of MPDR-R is initialized from the $(f_e, f_d)$ used in MPD. Even without a manifold ensemble, MPDR shows significant improvement over existing algorithms, including ones leveraging an autoencoder in EBM training [19, 24]. The performance of MPDR is stable over a range of $D_\mathbf{z}$, as demonstrated in Appendix B.1.

**MNIST OOD Detection**  To ensure that MPDR is not overfitted to the hold-out digit, we test MPDR in detecting five non-MNIST outlier datasets (Table 2). The results demonstrated that MPDR excels in detecting a wide range of outliers, surpassing the performance of naive algorithms such as autoencoders (AE) and scalar EBMs (IGEBM). Although MPDR achieves high overall detection performance, MPDR-R exhibits slightly weaker performance on EMNIST and Constant datasets. This can be attributed to the limited flexibility of the autoencoder-based energy function employed in MPDR-R.

**CIFAR-10 OOD Detection**  We evaluate MPDR on the CIFAR-10 inliers, a standard benchmark for EBM-based OOD detection. The manifold ensemble includes three convolutional autoencoders, with $D_\mathbf{z} = 32, 64, 128$. MPDR-S uses a ResNet energy function used in IGEBM [11]. MPDR-R adopts the ResNet-based autoencoder architecture used in NAE [24].

Table 3 compares MPDR to state-of-the-art EBMs. MPDR-R shows competitive performance across five OOD datasets, while MPDR-S also achieves high AUROC on SVHN and Constant. As both MPDR-R and NAE use the same autoencoder architecture for the energy, the discrepancy in performance can be attributed to the MPDR training algorithm. MPDR-R outperforms NAE on four out of five OOD datasets. Comparison between MPDR-S and DRL demonstrates the effectiveness of non-Gaussian manifold-aware perturbation used in MPDR. CLEL shows strong

Table 3: OOD detection with CIFAR-10 as in-distribution. AUROC values are presented. The largest value in the column is marked as boldface, and the second and the third largest values are underlined. Asterisks denote that the results are adopted from literature.

| | SVHN | Textures | Constant | CIFAR100 | CelebA |
|---|---|---|---|---|---|
| PixelCNN++ [31]* | 0.32 | 0.33 | 0.71 | 0.63 | - |
| GLOW [8]* | 0.24 | 0.27 | - | 0.55 | 0.57 |
| IGEBM [11]* | 0.63 | 0.48 | 0.39 | - | - |
| NVAE [21] | 0.4402 | 0.4554 | 0.6478 | 0.4943 | 0.6804 |
| VAEBM [18]* | 0.83 | - | - | 0.62 | 0.77 |
| JEM [32]* | 0.67 | 0.60 | - | 0.67 | 0.75 |
| Improved CD [14] | 0.7843 | 0.7275 | 0.8000 | 0.5298 | 0.5399 |
| NAE [24] | 0.9352 | 0.7472 | 0.9793 | 0.6316 | **0.8735** |
| DRL [16] | 0.8816 | 0.4465 | 0.9884 | 0.4377 | 0.6398 |
| CLEL [33]* | 0.9848 | **0.9437** | - | **0.7161** | 0.7717 |
| MPDR-S (ours) | **0.9860** | 0.6583 | **0.9996** | 0.5576 | 0.7313 |
| MPDR-R (ours) | 0.9807 | 0.7978 | **0.9996** | 0.6354 | 0.8282 |
| **MPDR-R Single AE** | | | | | |
| $D_\mathbf{z} = 32$ | 0.8271 | 0.6606 | 0.9877 | 0.5835 | 0.7751 |
| $D_\mathbf{z} = 64$ | 0.9330 | 0.6631 | 0.9489 | 0.6223 | 0.8272 |
| $D_\mathbf{z} = 128$ | 0.9886 | 0.6942 | 0.9651 | 0.6531 | 0.8500 |

Table 4: OOD detection on pretrained ViT-B_16 representation with CIFAR-100 as in-distribution. Performance is measured in AUROC.

| | CIFAR10 | SVHN | CelebA |
|---|---|---|---|
| **Supervised** | | | |
| MD [34] | 0.8634 | 0.9638 | 0.8833 |
| RMD [35] | 0.9159 | 0.9685 | 0.4971 |
| **Unsupervised** | | | |
| AE | 0.8580 | 0.9645 | 0.8103 |
| NAE [24] | 0.8041 | 0.9082 | 0.8181 |
| IGEBM [11] | 0.8217 | 0.9584 | 0.9004 |
| DRL [16] | 0.5730 | 0.6340 | 0.7293 |
| MPDR-S (ours) | 0.8338 | 0.9911 | **0.9183** |
| MPDR-R (ours) | **0.8626** | **0.9932** | 0.8662 |
| **MPDR-R Single AE** | | | |
| $D_\mathbf{z} = 128$ | 0.6965 | 0.9326 | 0.9526 |
| $D_\mathbf{z} = 256$ | 0.8048 | 0.9196 | 0.7772 |
| $D_\mathbf{z} = 1024$ | 0.7443 | 0.9482 | 0.9247 |

overall performance, indicating that learning semantic information is important in this benchmark. Incorporating contrastive learning into MPDR framework is an interesting future direction.

**CIFAR-100 OOD Detection on Pretrained Representation** In Table 4, we test MPDR on OOD detection with CIFAR-100 inliers. To model a distribution of diverse images like CIFAR-100, we follow [34] and apply generative modeling in the representation space from a large-scale pretrained model. As we assume an unsupervised setting, we use pretrained representations without fine-tuning. Input images are transformed into 768D vectors by ViT-B_16 model [36]. ViT outputs are normalized with its norm and projected onto a hypersphere. We observe that adding a small Gaussian noise of 0.01 to training data improves stability of all algorithms. We use MLP for all energy functions and autoencoders. In MPDR, the manifold ensemble comprises three autoencoders with $D_\mathbf{z} = 128, 256, 1024$. We also implement supervised baselines (MD [34] and RMD [35]). The spherical projection is not applied for MD and RMD to respect their original implementation.

MPDR demonstrates strong anomaly detection performance in the representation space, with MPDR-S and MPDR-R outperforming IGEBM and AE/NAE, respectively. This success can be attributed to the low-dimensional structure often found in the representation space of in-distribution data, as observed in [37]. MPDR's average performance is nearly on par with supervised methods, MD and RMD, which utilize class information. Note that EBM inputs are no longer images, making previous EBM training techniques based on image transformation [14, 33] inapplicable.

**Ablation Study** Table 3 and 4 also report the results from single-manifold MPDR-R with varying latent dimensionality $D_\mathbf{z}$ to show MPDR's sensitivity to a choice of an autoencoder manifold. Manifold ensemble effectively hedges the risk of relying on a single autoencoder which may not be optimal for detecting all types of outliers. Furthermore, manifold ensemble often achieves better AUROC score than MPDR with a single autoencoder. Additional ablation studies can be found in the Appendix. In Sec. B.2, we examine the sensitivity of MPDR to noise magnitude $\sigma$ and the effectiveness of the noise magnitude ensemble. Also in Sec. B.4.4, we investigate the effect to scaling parameter $\gamma$ and show that the training is unstable when $\gamma$ is too large.

**Multi-Class Anomaly Detection on MVTec-AD Using Pretrained Representation** MVTec-AD [38] is also a popular anomaly detection benchmark dataset, containing images of 15 objects in their normal and defective forms. We follow the "unified" experimental setting from [39]. Normal images from 15 classes are used as the training set, where no label information is provided. We use the same feature extraction procedure used in [39]. Each image is transformed to a $272 \times 14 \times 14$ feature map using EfficientNet-b4. When running MPDR, we treat each spatial dimension of a feature map

Table 5: Acoustic anomaly detection on DCASE 2020 Track 2 Dataset. AUROC and pAUROC (in parentheses) are displayed per cent. pAUROC is defined as AUROC computed over a restricted false positive rate interval $[0, p]$, where we set $p = 0.1$, following [41].

|  | Toy Car | Toy Conveyor | Fan | Pump | Slider | Valve |
|---|---|---|---|---|---|---|
| AE [41] | 75.40 (62.03) | 77.38 (63.02) | 66.44 (53.40) | 71.42 (61.77) | 89.65 (74.69) | 72.52 (**52.02**) |
| MPDR-R | **81.54 (68.21)** | **78.61 (63.99)** | **71.72 (55.95)** | **78.27 (68.14)** | **90.91 (76.58)** | **75.23** (51.04) |
| IDNN [44] | 76.15 (72.36) | 78.87 (62.50) | 72.74 (54.30) | 73.15 (61.25) | 90.83 (74.16) | 90.27 (69.46) |
| MPDR-IDNN | **78.53 (73.34)** | **79.54 (65.35)** | **73.27 (54.57)** | **76.58 (66.49)** | **91.56 (75.19)** | **91.10 (70.87)** |

as an input vector to MPDR, transforming the task into a 272D density estimation problem. We normalize a 272D vector with its norm and add a small white noise with a standard deviation of 0.01 during training. We use the maximum energy value among 14×14 vectors as an anomaly score of an image. For the localization task, we resize 14×14 anomaly score map to 224x224 image and compute AUROC for each pixel with respect to the ground true mask. We compare MPDR-R with UniAD [39] and DRAEM [40]. The results are shown in Table 12. Details on MPDR implementation can be found in B.5.

MPDR achieves AUROC scores that are very close to that of UniAD, outperforming DRAEM. The trend is consistent in both detection and localization tasks. The performance gap between MPDR and UniAD can be attributed to the fact that UniAD leverages spatial information of 14×14 feature map while our implementation of MPDR processes each pixel in the feature map separately.

## 4.4 Acoustic Anomaly Detection

We apply MPDR to anomaly detection with acoustic signals, another form of non-image data. We use DCASE 2020 Challenge Task 2 dataset [41], containing recordings from six different machines, with three to four instances per machine type. The task is to detect anomalous sounds from deliberately damaged machines, which are unavailable during training. We applied the standard preprocessing methods in[41] to obtain a 640-dimensional vector built up of consecutive mel-spectrogram features. Many challenge submissions exploit dataset-specific heuristics and ensembles for high performance, e.g., [42, 43]. Rather than competing, we focus on demonstrating MPDR's effectiveness in improving common approaches, such as autoencoders and Interpolation Deep Neural Networks (IDNN) [44]. IDNN is a variant of masked autoencoders which predicts the middle (the third) frame given the remaining frames. Similarly to autoencoders, IDNN predicts an input as anomaly when the prediction error is large. We first train AE and IDNN for 100 epochs and then apply MPDR by treating the reconstruction (or prediction) error as the energy. Manifold ensemble consists of autoencoders with $D_{\mathbf{z}} = 32, 64, 128$. More training details can be found in the Appendix.

Table 5 shows that MPDR improves anomaly detection performance for both AE and IDNN. The known failure mode of AE and IDNN is producing unexpected low prediction (or reconstruction) error for anomalous inputs. By treating the error as the energy and applying generative training, the error in OOD region is increased through training, resulting in improved anomaly detection performance.

## 4.5 Anomaly Detection on Tabular Data

We test MPDR on ADBench [45], which consists 47 tabular datasets for anomaly detection. We compare MPDR with 13 baselines from ADBench. The baseline results are reproduced using the official ADBench repository. We consider the setting where the training split does not contain anomalies. For each dataset, we run each algorithm on three random splits and computed the AUROC on the corresponding test split. We then ranked the algorithms based on the averaged AUROC. In Table A, we present a summary table with the average rank across the 47 datasets.

We employ MPDR-R with a single manifold. For both the manifold and the energy, the same MLP-based autoencoder architecture is used. The encoder and the decoder are MLPs with two 1024-hidden-neuron layers. If the input space dimensionality is smaller than or equal to 100, the latent space dimensionality is set to have the same dimensionality $D_{\mathbf{z}} = D_{\mathbf{x}}$. If $D_{\mathbf{x}} > 100$, we set $D_{\mathbf{z}}$ as 70% of $D_{\mathbf{x}}$. We employ 1 step of Langevin Monte Carlo in the latent space and 5 steps

in the input space. The step sizes are 0.1 for the latent space and 10 for the input space. All the hyperparameters except $D_{\mathbf{z}}$ are fixed across 47 datasets.

As shown in Table 6, MPDR achieves highly competitive performance on ADBench, demonstrating a higher average rank than the isolation forest, with some overlap of confidence interval. This result indicates that MPDR is a general-purpose anomaly detection algorithm capable of handling tabular data. We empirically observe that MPDR is most effective when AUROC from its autoencoder is low, meaning that the outliers are near the autoencoder manifold. When the autoencoder already achieves good AUROC, the improvement from MPDR training is often marginal.

## 5   Related Work

Leveraging autoencoders in training, MPDR is closely related to other EBM training algorithms that incorporate auxiliary modules. While the use of a single variational autoencoder in EBM training is explored in [18, 19, 46], MPDR employs multiple autoencoders. MPDR is also compatible with non-variational autoencoders, offering greater flexibility. Normalizing flows [17, 25, 47] provide informative negative samples and a latent space for EBM training, but unlike MPDR, they do not exploit the low-dimensional structure of data. EBMs can also be trained with additional generator modules [48, 49], which plays a similar role to the decoder of an autoencoder. A contrastive representation learning module [33] improves EBM performance but relies on domain-specific data augmentations and is only demonstrated on images. In contrast, MPDR is applicable to a wider range of data, utilizing a more general assumption of low-dimensionality in data.

MPDR presents a novel objective function for EBM by extending recovery likelihood framework [22, 16]. Investigating its connection to other previously studied objective functions, such as $f$-divergence [50], pseudo-spherical scoring rule [51], divergence triangle [46], and Stein discrepancy [52], would also be an interesting future direction.

MPDR contributes to the field of anomaly detection [45, 39, 40, 53, 54]. The promising results from MPDR demonstrates that learning the distribution of data is a principled and effective approach for detecting anomalies [24].

Table 6: The rank of AUROC averaged over 47 datasets in ADBench. A smaller value indicates the algorithm achieves higher AUROC score compared to other algorithms on average. The standard errors are also displayed.

| Method | Average Rank |
|---|---|
| MPDR (ours) | **4.43** $\pm$ 0.50 |
| IForest | 5.28 $\pm$ 0.39 |
| OCSVM | 7.94 $\pm$ 0.47 |
| CBLOF | 5.98 $\pm$ 0.53 |
| COF | 9.23 $\pm$ 0.58 |
| COPOD | 7.10 $\pm$ 0.61 |
| ECOD | 6.97 $\pm$ 0.58 |
| HBOS | 6.53 $\pm$ 0.51 |
| KNN | 6.64 $\pm$ 0.56 |
| LOF | 9.04 $\pm$ 0.62 |
| PCA | 6.45 $\pm$ 0.64 |
| SOD | 7.91 $\pm$ 0.52 |
| DeepSVDD | 11.43 $\pm$ 0.42 |
| DAGMM | 10.09 $\pm$ 0.52 |

## 6   Conclusion

**Contributions**   We propose MPDR, a novel method of utilizing autoencoders for training EBM. An autoencoder in MPDR provides an informative starting point for MCMC, offers the latent space for the effective traversal of a high-dimensional space, and guides the drift of an MCMC sampler (Eq. 7). MPDR performs competitively on various anomaly detection benchmarks involving diverse types of data, contributing to the enhancement of generative modeling for anomaly detection with high-dimensional data.

**Limitations**   First, the practical performance of MPDR is still sensitive to the specifics of autoencoders used in MPD. Second, some data, such as high-resolution images or texts, may not exhibit a clear low-dimensional structure. In these cases, MPDR may require a separate representation learning stage, as demonstrated in our CIFAR-100 experiment with ViT (Sec. 4.3). Third, MPDR is not optimized for generating samples starting from a simple distribution, such as a Gaussian, while DRL is. We may resort to longer-chain MCMC to generate samples from MPDR.

## Acknowledgments and Disclosure of Funding

S. Yoon would like to thank Luigi Favaro and Tilman Plehn for their helpful discussions during the early development of the algorithm. S. Yoon is supported by a KIAS Individual Grant (AP095701) via the Center for AI and Natural Sciences at Korea Institute for Advanced Study. Y.-K. Noh was partly supported by NRF/MSIT (No. 2018R1A5A7059549, 2021M3E5D2A01019545) and IITP/MSIT (IITP-2021-0-02068, 2020-0-01373, RS-2023-00220628). This work was supported in part by IITP-MSIT grant 2021-0-02068 (SNU AI Innovation Hub), IITP-MSIT grant 2022-0-00480 (Training and Inference Methods for Goal-Oriented AI Agents), KIAT grant P0020536 (HRD Program for Industrial Innovation), ATC+ MOTIE Technology Innovation Program grant 20008547, SRRC NRF grant RS-2023-00208052, SNU-AIIS, SNU-IAMD, SNU BK21+ Program in Mechanical Engineering, and SNU Institute for Engineering Research.

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

# A  Consistency of MPDR

Let us denote our model for recovery likelihood as $p(\mathbf{x}|\tilde{\mathbf{z}}; \theta)$. We assume that this model is identifiable (different model parameters correspond to different distribution) and correctly specified (there exists $\theta^*$ such that $p(\mathbf{x}|\tilde{\mathbf{z}}; \theta^*) = p_{data}(\mathbf{x}|\tilde{\mathbf{z}})$). We also assume that $p(\mathbf{x}|\tilde{\mathbf{z}}; \theta)$ is non-zero for all $\mathbf{x}$, $\tilde{\mathbf{z}}$, and $\theta$. Our objective is to $\max_\theta \frac{1}{N} \sum_{i=1}^N \log p(\mathbf{x}_i|\tilde{\mathbf{z}}_i; \theta)$ where $(\mathbf{x}_i, \tilde{\mathbf{z}}_i) \sim p(\mathbf{x}, \tilde{\mathbf{z}}) = p_{data}(\mathbf{x})p(\tilde{\mathbf{z}}|\mathbf{x})$. In the limit of $N \to \infty$, the average converges to the expectation $\frac{1}{N} \sum_{i=1}^N \log p(\mathbf{x}_i|\tilde{\mathbf{z}}_i; \theta) \to \mathbb{E}_{(\mathbf{x},\tilde{\mathbf{x}})}[\log p(\mathbf{x}|\tilde{\mathbf{z}}; \theta)]$. If we subtract $\mathbb{E}_{(\mathbf{x},\tilde{\mathbf{x}})}[\log p(\mathbf{x}|\tilde{\mathbf{z}}; \theta^*)]$ which is constant with respect to $\theta$, then the expression can be written with respect to KL divergence as follows:

$$\mathbb{E}_{(\mathbf{x},\tilde{\mathbf{x}})}[\log p(\mathbf{x}|\tilde{\mathbf{z}}; \theta) - \log p(\mathbf{x}|\tilde{\mathbf{z}}; \theta^*)] = \int p(\mathbf{x}, \tilde{\mathbf{z}}) \log \frac{p(\mathbf{x}|\tilde{\mathbf{z}}; \theta)}{p(\mathbf{x}|\tilde{\mathbf{z}}; \theta^*)} d\mathbf{x} d\tilde{\mathbf{z}} \tag{9}$$

$$= -\int p(\tilde{\mathbf{z}}) \, \mathrm{KL}(p(\mathbf{x}|\tilde{\mathbf{z}}; \theta^*)||p(\mathbf{x}|\tilde{\mathbf{z}}; \theta)) d\tilde{\mathbf{z}} \tag{10}$$

The maximum of Eq. 10 is 0, as the minimum of KL divergence is 0. The maximum is achieved if and only if $\theta = \theta^*$. Note that $p(\tilde{\mathbf{z}})$ is assumed to be constant with respect to $\theta$.

Since we did not rely on the specifics of how a perturbation $p(\tilde{\mathbf{z}}|\mathbf{x})$ is actually performed, this consistency result holds for any choices of the encoder $f_e$ and the noise magnitude $\sigma$, as long as the recovery likelihood $p(\mathbf{x}|\tilde{\mathbf{z}}; \theta)$ remains non-zero for all $\mathbf{x}$, $\tilde{\mathbf{z}}$, and $\theta$.

# B  Experimental Details and Additional Results

In this section, we provide detailed information on how each experiment is conducted and also provide some additional supporting experimental results. The hyperparameters related to LMC is summarized in Table 7. ConvNet architectures are provided in Table 8. The contents are organized by the training dataset.

Table 7: Hyperparameters for LMC. Latent chain hyperparameters are denoted by $\mathcal{Z}$ and $\mathcal{X}$ indicates visible chain hyperparameters. "scale ($\gamma$)" refers to the multiplicative scale factor on the perturbation probability.

| Experiment | $\mathcal{Z}$ Steps | $\mathcal{Z}$ Step Size | $\mathcal{Z}$ Noise | $\mathcal{Z}$ scale ($\gamma$) | $\mathcal{X}$ Steps | $\mathcal{X}$ Step Size | $\mathcal{X}$ Noise | $\mathcal{X}$ scale ($\gamma$) |
|---|---|---|---|---|---|---|---|---|
| **MNIST** | | | | | | | | |
| MPDR-S | 2 | 0.05 | 0.02 | 0.0001 | 5 | 10 | 0.005 | 0 |
| MPDR-R | 5 | 0.1 | 0.02 | 0.0001 | 5 | 10 | 0.005 | 0 |
| **CIFAR-10** | | | | | | | | |
| MPDR-S | 10 | 0.1 | 0.01 | 0.0001 | 20 | 10 | 0.005 | 0 |
| MPDR-R | 10 | 0.1 | 0.01 | 0.0001 | 20 | 10 | 0.005 | 0 |
| **CIFAR-100 + ViT** | | | | | | | | |
| MPDR-S | 0 | - | - | - | 30 | 1 | 0.005 | 0.0001 |
| MPDR-R | 0 | - | - | - | 30 | 1 | 0.005 | 0.0001 |
| **DCASE 2020** (Toy Car, Toy Conveyor, Pump) | | | | | | | | |
| MPDR-R | 0 | - | - | - | 5 | 10 | 0.005 | 0.0001 |
| MPDR-IDNN | 0 | - | - | - | 5 | 10 | 0.005 | 0.0001 |
| **DCASE 2020** (Fan, Slider, Valve) | | | | | | | | |
| MPDR-R | 0 | - | - | - | 5 | 10 | 0.005 | 0.0001 |
| MPDR-IDNN | 20 | 0.1 | 0.01 | 0.0001 | 20 | 10 | 0.005 | 0.0001 |
| **MVTec-AD** | | | | | | | | |
| MPDR-R | 0 | - | - | - | 10 | 0.1 | 0.1 | 0.0001 |
| **ADBench** | | | | | | | | |
| MPDR-R | 1 | 0.1 | 0.05 | 0.0001 | 5 | 10 | 0.1 | 0.0001 |

### B.1 MNIST

#### B.1.1 Datasets

All input images are 28×28 grayscale, and each pixel value is represented as a floating number between $[0, 1]$. Models are trained on the training split of MNIST[2], excluding the digit designated to be held-out. The training split contains 60,000 images, and the hold-out procedure reduces the training set to approximately 90%. We evaluate the models on the test split of MNIST, which contains a total of 10,000 images. For non-MNIST datasets used in evaluation, we only use their test split. All non-MNIST datasets used in the experiment also follow the 28×28 grayscale format, similar to MNIST.

- KMNIST (KMNIST-MNIST) [55] [3] contains Japanese handwritten letters, pre-processed into the same format as MNIST. The license of KMNIST is CC BY-SA 4.0.
- EMNIST (EMNIST-Letters) [56] contains grayscale handwritten English alphabet images. Its test split contains 20,800 samples. No license information is available.
- For the Omniglot [57] dataset, the evaluation set is used. The set consists of 13,180 images. No license information is available.
- We use the test split of FashionMNIST [58], which contains 10,000 images. The dataset is made public under the MIT license.
- Constant dataset is a synthetic dataset that contains 28×28 images, where all pixels have the same value. The value is randomly drawn from a uniform distribution over $[0, 1]$. We use 4,000 constant images.

#### B.1.2 Autoencoder Implementation and Training

The encoder and the decoder used in MPDR, $f_e$ and $f_d$, have the architecture of MNIST Encoder and MNIST Decoder, provided in Table 8, respectively. We use the spherical latent space $\mathbb{S}^{D_{\mathbf{z}}-1}$ where $D_{\mathbf{z}} = 32$ for the main experiment. The autoencoder is trained to minimize the $l_2$ reconstruction error of the training data for 30 epochs with Adam of learning rate $1 \times 10^{-4}$. The batch size is 128 and no data augmentation is applied. The $l_2$ norm of the encoder is regularized with the coefficient of $1 \times 10^{-4}$. The same autoencoder is used for both MPDR-S and MPDR-R.

#### B.1.3 MPDR Implementation and Training

In MPDR-S, the energy function $E_\theta$ has the architecture of MNIST Encoder (Table 8) with $D_{\mathbf{z}} = 1$. The network is randomly initialized from PyTorch default setting and the spectral normalization is applied. The energy function is trained with MPDR algorithm for 50 epochs. The learning rate is $1 \times 10^{-4}$. The batch size is 128. The perturbation probability scaling factor $\gamma$ for the visible LMC chain is set to zero. For an image-like data such as MNIST, the gradient of perturbation probability $\nabla_{\mathbf{x}}(||\tilde{\mathbf{z}} - f_e(\mathbf{x})||^2)$ introduces non-smooth high-frequency patterns resembling adversarial noise, harming the stability of the training. Therefore, we only use non-zero $\gamma$ in latent chains in MNIST and CIFAR-10 experiments.

For MPDR-R, the energy function is initialized from $(f_e, f_d)$ and then trained through MPDR algorithm. The learning rate is set to $1 \times 10^{-5}$. All the other details are identical to the MPDR-S case.

### B.2 Sensitivity to $\sigma$

As an ablation study for noise magnitude ensemble, we perform single-noise-magnitude experiment for MPDR and examine MPDR's sensitivity to the noise magnitude $\sigma$. We evaluate the OOD detection performance of MPDR-S on the MNIST hold-out digit 9 setting with varying values of $\sigma$. Results are shown in Table 9.

The choice of $\sigma$ has a significant impact on MPDR's OOD detection performance. In Table 9, $\sigma = 0.01$ gives poor OOD detection performance, particularly with respect to the hold-out digit. The performance generally improves as $\sigma$ grows larger, but a large $\sigma$ is not optimal for detecting EMNIST.

---

[2]http://yann.lecun.com/exdb/mnist/
[3]https://github.com/rois-codh/kmnist

Table 8: Convolutional neural network architectures used in experiments. The parenthesis following the network name indicates the activation function used in the network.

| MNIST Encoder (ReLU) | MNIST Decoder (ReLU) |
| --- | --- |
| $Conv_3(1, 32)$ | $ConvT_4(D_\mathbf{z}, 128)$ |
| $Conv_3(32, 64)$ | Upsample(2x) |
| MaxPool(2x) | $ConvT_3(128, 64)$ |
| $Conv_3(64, 64)$ | $ConvT_3(64, 64)$ |
| $Conv_3(64, 128)$ | Upsample(2x) |
| MaxPool(2x) | $ConvT_3(64, 32)$ |
| $Conv_4(128, 1024)$ | $ConvT_3(32, 1)$ |
| $FC(1024, D_\mathbf{z})$ | Sigmoid() |

| CIFAR-10 Encoder 1 | CIFAR-10 Decoder 1 |
| --- | --- |
| $Conv_4(3, 32, stride=2)$ | $ConvT_8(D_\mathbf{z}, 256)$ |
| $Conv_4(32, 64, stride=2)$ | $ConvT_4(256, 128, stride=2, pad=1)$ |
| $Conv_4(64, 128, stride=2)$ | $ConvT_4(128, 64, stride=2, pad=1)$ |
| $Conv_2(128, 256, stride=2)$ | $ConvT_1(64, 3)$ |
| $FC(256, D_\mathbf{z})$ | Sigmoid() |

| CIFAR-10 Encoder 2 | |
| --- | --- |
| $Conv_3(3, 128, pad=1)$ | |
| ResBlock(128, 128, down=True) | CIFAR-10 Decoder 2 |
| ResBlock(128, 128) | |
| ResBlock(128, 256, down=True) | $ConvT_4(D_\mathbf{z}, 128)$ |
| ResBlock(256, 256) | ResBlock(128, 128, up=True) |
| ResBlock(256, 256, down=True) | ResBlock(128, 128, up=True) |
| ResBlock(256, 256) | ResBlock(128, 128, up=True) |
| GlobalAvgPool() | $Conv_3(128, 3, pad=1)$ |
| $FC(256, D_\mathbf{z})$ | |

Selecting a single optimal $\sigma$ will be very difficult, and therefore, we employ noise magnitude ensemble which can hedge the risk of choosing a suboptimal value for $\sigma$.

### B.2.1 Sensitivity to $D_\mathbf{z}$

As an ablation study for manifold ensemble, we investigate the sensitivity of MPDR to the choice of the latent space dimensionality of the autoencoder. We evaluate the OOD detection performance of MPDR-S on the MNIST hold-out digit 9 setting with varying values of $D_\mathbf{z}$. Table 10 presents the results. Consequently, MPDR runs stably for a large range of $D_\mathbf{z}$, producing decent OOD performance. One hypothesis is that, for MNIST, it is relatively easy for these autoencoders to capture the manifold structure of MNIST sufficiently well.

Meanwhile, we do observe that the choice of $D_\mathbf{z}$ affects OOD performance in an interesting way. Increasing $D_\mathbf{z}$ enhances AUPR for certain OOD datasets but deteriorates AUPR for others. For example, AUPR of Omniglot is increased with larger $D_\mathbf{z}$, but AUPR of EMNIST, FashionMNIST, and Constant dataset decreases. No single autoencoder is optimal for detecting all outlier datasets. This observation motivates the use of manifold ensemble, employed in non-MNIST MPDR experiments.

### B.2.2 Note on Reproduction

For an autoencoder-based outlier detector, denoted as "AE" in Table 1, we use the same autoencoder used in MPDR with $D_\mathbf{z}$.

NAE is reproduced based on its public code base[4].

We tried to reproduce DRL on MNIST but failed to train it stably. The original paper and the official code base also does not provide a guidance on training DRL on MNIST.

### B.3 CIFAR-10 OOD Detection

### B.3.1 Datasets

All data used in CIFAR-10 experiment are in the $32 \times 32$ RGB format. Models are only trained on CIFAR-10 training set, and evaluated on the testing set of each dataset.

- CIFAR-10 [59] contains 60,000 training images and 10,000 testing images. Models are trained on the training set. We don't use its class information, as we consider only unsupervised setting. No license information available.

---

[4]https://github.com/swyoon/normalized-autoencoders

Table 9: Sensitivity to $\sigma$. MPDR-S is run with an autoencoder with varying values of noise magnitude $\sigma$. AUPR against various outlier datasets are presented. For MNIST 9, we present the standard deviation computed over the last 10 epochs.

| $\sigma$ | MNIST 9 | KMNIST | EMNIST | Omniglot | FashionMNIST | Constant |
|---|---|---|---|---|---|---|
| 0.01 | $0.098 \pm 0.009$ | 0.667 | 0.827 | 0.940 | 0.944 | 0.895 |
| 0.1 | $0.330 \pm 0.063$ | 0.983 | 0.995 | 0.971 | 0.994 | 0.998 |
| 0.2 | $0.522 \pm 0.053$ | 0.999 | 0.993 | 0.997 | 0.999 | 1.000 |
| 0.3 | $0.558 \pm 0.039$ | 1.000 | 0.990 | 0.997 | 1.000 | 1.000 |

Table 10: Sensitivity to $D_{\mathbf{z}}$. MPDR-S is run with an autoencoder with varying values of $D_{\mathbf{z}}$. AUPR against various outlier datasets are presented. For MNIST 9, we present the standard deviation computed over the last 10 epochs. Noise magnitude ensemble is applied.

| $D_{\mathbf{z}}$ | MNIST 9 | KMNIST | EMNIST | Omniglot | FashionMNIST | Constant |
|---|---|---|---|---|---|---|
| 16 | $0.611 \pm 0.041$ | 0.979 | 0.996 | 0.958 | 0.999 | 1.000 |
| 32 | $0.525 \pm 0.039$ | 0.999 | 0.994 | 0.994 | 0.999 | 1.000 |
| 64 | $0.512 \pm 0.048$ | 0.999 | 0.993 | 0.998 | 0.998 | 0.999 |
| 128 | $0.505 \pm 0.051$ | 0.999 | 0.991 | 0.999 | 0.988 | 0.946 |
| 256 | $0.590 \pm 0.041$ | 0.996 | 0.983 | 0.996 | 0.958 | 0.866 |

- SVHN [60] is a set of digit images. Its test set containing 26,032 is used in the experiment. The dataset is non-commercial use only.
- Texture [61] dataset, also called Describable Textures Dataset (DTD), contains 1,880 test images. The images are resized into 32×32. No license information available.
- CelebA [62][5] is a dataset of cropped and aligned human face images. The test set contains 19,962 images. The dataset is for non-commercial research purposes only. We center-crop each image into $140 \times 140$ and then resize it into $32 \times 32$.
- Constant dataset is a synthetic dataset that contains 4,000 32×32 RGB monochrome image. All 32×32 pixels have the same RGB value which is drawn uniform-randomly from $[0, 1]^3$.
- CIFAR-100 [59] contains 60,000 training images and 10,000 testing images. No license information available.

### B.3.2 Autoencoder Implementation and Training

The autoencoders for CIFAR-10 experiment have an architecture of "CIFAR-10 Encoder 1" and "CIFAR-10 Decoder 1" in Table 8 with $D_{\mathbf{z}} = 32, 64, 128$. Each autoencoder is trained for 40 epoch with learning rate $1 \times 10^{-4}$ and batch size 128. During training the autoencoders, we apply the following data augmentation operations: random horizontal flipping with the probability of 0.5, random resize crop with the scale parameter $[0.08, 1]$ with the probability 0.2, color jittering with probability of 0.2, random grayscale operation with the probability 0.2. The color jittering parameters are the same with the one used in SimCLR [63] (brightness 0.8, contrast 0.8, saturation 0.8, hue 0.4, with respect to `torchvision.transorms.ColorJitter` implementation). The $l_2$ norm of an encoder is regularized with the coefficient of 0.00001. The same autoencoder is used for both MPDR-S and MPDR-R. To boost the performance, we heuristically introduce sample replay buffer for negative samples and apply data augmentation in the middle of LMC.

### B.3.3 MPDR Implementation and Training

The energy function in MPDR-S is "CIFAR-10 Encoder 2" with $D_{\mathbf{z}} = 1$. The energy function in MPDR-R is "CIFAR-10 Encoder 2" and "CIFAR-10 Decoder 2" with $D_{\mathbf{z}} = 1$. In MPDR-R, the energy function is pre-trained by minimizing the reconstruction error of the training data for 40 epochs. Only random horizontal flip is applied and no other data augmentation is used. Similarly to the MNIST case, we set $\gamma = 0$ for the visible LMC chain.

---

[5]`https://mmlab.ie.cuhk.edu.hk/projects/CelebA.html`

### B.3.4 Note on Reproduction

For Improved CD, we train a CIFAR-10 model from scratch using the training script provided by the authors without any modification [6]. We use the model with the best Inception Score to compute AUROC scores.

For DRL, we use the official checkpoint for T6 CIFAR-10 model [7] and compute its energy to perform OOD detection.

Also for NVAE, we use the official CIFAR-10 checkpoint provided in the official repository [8]. We use negative log-likelihood as the outlier score. Due to computational budget constraint, we could only set the number of importance weighted sample to be one `-num_iw_samples=1`.

As in MNIST, we use the official CIFAR-10 checkpoint of NAE provided by the authors.

## B.4 CIFAR-100 OOD Detection on Pretrained Representation

### B.4.1 Datasets

CIFAR-100, CIFAR-10, SVHN, and CelebA datasets are used and are described in the previous section. Each image is resized to $224 \times 224$ and fed to ViT-B_16 to produce a 768-dimensional vector. MD and RMD operate with this vector. For other methods, the 768D vector is projected onto a hypersphere.

### B.4.2 Autoencoder Implementation and Training

Each encoder and decoder is an MLP with two hidden layers where each layer contains 1024 hidden neurons. The leaky ReLU activation function is used in all hidden layers. We use $D_{\mathbf{z}} = 128, 256, 1024$. The autoencoders are trained to minimize the reconstruction error of the training data. During training, the Gaussian noise with the standard deviation of 0.01 is added to each training sample. The $l_2$ norm of the encoder's weights are regularized with the coefficient of $1 \times 10^{-6}$.

### B.4.3 MPDR Implementation and Training

The energy functions are also MLPs. The energy function of MPDR-S has the same architecture as the encoder of the autoencoder with $D_{\mathbf{z}} = 1$. The energy function of MPDR-R is an autoencoder with the latent dimensionality of 1024. The energy functions are trained for 30 epochs with the learning rate of $1 \times 10^{-4}$.

### B.4.4 Sensitivity to $\gamma$

We examine how $\gamma$ affects MPDR's performance. As seen in Table 11, MPDR shows the best performance on the small $\gamma$ regime, roughly from 0.0001 to 0.001. Setting too large $\gamma$ is detrimental for the performance and often even incurs training instabilities. It is interesting to note that $\gamma = 0$ also gives a decent result. One possible explanation is that $\gamma = 0$ reduces the negative sample distribution $p_\theta(\mathbf{x}|\tilde{\mathbf{z}})$ to the model distribution $p_\theta(\mathbf{x})$, which is still a valid negative sample distribution for training an EBM.

Table 11: Sensitivity of $\gamma$, demonstrated in CIFAR-100 experiment. AUROC values are displayed.

| $\gamma$ | CIFAR10 | SVHN | CelebA |
|---|---|---|---|
| 0 | 0.8580 | 0.9931 | 0.8456 |
| 0.0001 | 0.8626 | 0.9932 | 0.8662 |
| 0.001 | 0.8639 | 0.9918 | 0.8625 |
| 0.01 | 0.8496 | 0.9894 | 0.8576 |
| 0.1 | 0.8186 | 0.9424 | 0.8511 |

---

[6]`https://github.com/yilundu/improved_contrastive_divergence`
[7]`https://github.com/ruiqigao/recovery_likelihood`
[8]`https://github.com/NVlabs/NVAE`

Table 12: MVTec-AD detection and localization task in the unified setting. AUROC scores (percent) are computed for each class. UniAD and DRAEM results are adopted from [39]. The largest value in a task is marked as boldface.

| | Detection | | | Localization | | |
| | MPDR (ours) | UniAd[39] | DRAEM [40] | MPDR (ours) | UniAd[39] | DRAEM[40] |
| --- | --- | --- | --- | --- | --- | --- |
| Bottle | **100.0** | 99.7 | 97.5 | **98.5** | 98.1 | 87.6 |
| Cable | **95.5** | 95.2 | 57.8 | 95.6 | **97.3** | 71.3 |
| Capsule | 86.4 | **86.9** | 65.3 | 98.2 | **98.5** | 50.5 |
| Hazelnut | **99.9** | 99.8 | 93.7 | **98.4** | 98.1 | 96.9 |
| Metal Nut | **99.9** | 99.2 | 72.8 | 94.5 | **94.8** | 62.2 |
| Pill | **94.0** | 93.7 | 82.2 | 94.9 | **95.0** | 94.4 |
| Screw | 85.9 | **87.5** | 92.0 | 98.1 | **98.3** | 95.5 |
| Toothbrush | 89.6 | **94.2** | 90.6 | **98.7** | 98.4 | 97.7 |
| Transistor | 98.3 | **99.8** | 74.8 | 95.4 | **97.9** | 65.5 |
| Zipper | 95.3 | 95.8 | **98.8** | 96.2 | 96.8 | **98.3** |
| Carpet | **99.9** | 99.8 | 98.0 | **98.8** | 98.5 | 98.6 |
| Grid | 97.9 | 98.2 | **99.3** | **96.9** | 96.5 | 98.7 |
| Leather | **100.0** | **100** | 98.7 | 98.5 | **98.8** | 97.3 |
| Tile | **100.0** | 99.3 | 98.7 | 94.6 | 91.8 | **98.0** |
| Wood | 97.9 | 98.6 | **99.8** | 93.8 | 93.2 | **96.0** |
| Mean | 96.0 | **96.5** | 88.1 | 96.7 | **96.8** | 87.2 |

## B.5 MVTec-AD Experiment

We use MPDR-R with a single manifold (i.e., without the manifold ensemble). Both the manifold and the energy are a fully connected autoencoder of 256D spherical latent space. For input-space Langevin Monte Carlo, the number of MCMC steps, the step size, and the noise standard deviation are 10, 0.1, and 0.1, respectively. No latent chain is used. The manifold is trained for 200 epochs with Adam of learning rate 1e-3, and the energy is trained for 20 epochs with Adam of learning rate 1e-4.

## B.6 Acoustic Anomaly Detection

### B.6.1 Dataset

The dataset consists of audio recordings with a duration of approximately 10 seconds, obtained through a single channel and downsampled to 16kHz. Each recording includes both the operational sounds of the target machine and background noise from the surrounding environment. The addition of noise was intended to replicate the conditions of real-world inspections, which often occur in noisy factory environments. The dataset covers six types of machinery, including four sampled from the MIMII Dataset (i.e., valve, pump, fan, and slide rail) and two from the ToyADMOS dataset (i.e., toy-car and toy-conveyor).

### B.6.2 Preprocessing

We follow the standard preprocessing scheme used in the challenge baseline and many of challenge participants. The approach involves the use of Short Time Fourier Transform (STFT) to transform each audio clip into a spectrogram, which is then converted to Mel-scale. We set the number of Mel bands as 128, the STFT window length as 1024, and the hop length (i.e., the number of audio samples between adjacent STFT columns) as 512. This configuration results in a spectrogram with 128 columns that represented the number of Mel bands. To construct the final spectrogram, the mel spectra of five consecutive frames are collected and combined to form a single row, resulting in a spectrogram with 640 columns. Each row of the spectrogram is sampled and used as input to the models under investigation, with a batch size of 512, meaning that 512 rows were randomly selected from the spectrogram at each iteration. We standardize all data along the feature dimension to zero mean and unit variance.

### B.6.3 Performance Measure

We refer to the scoring method introduced in [41] and use the area under the receiver operating characteristic curve (AUROC) and partial-AUROC (*pAUROC*) as a quantitative measure of performance. pAUROC measures the AUC over the area corresponding to the false positive rate from 0 to a reasonably small value $p$, which we set in all our experiments as 0.1. Each measure is defined as follows:

$$\text{AUROC} = \frac{1}{N_- N_+} \sum_{i=1}^{N_-} \sum_{i=j}^{N_+} \mathcal{H}(\mathcal{A}_\theta(x_j^+) - \mathcal{A}_\theta(x_i^-)) \tag{11}$$

$$pAUROC = \frac{1}{\lfloor pN_- \rfloor N_+} \sum_{i=1}^{\lfloor pN_- \rfloor} \sum_{i=j}^{N_+} \mathcal{H}(\mathcal{A}_\theta(x_j^+) - \mathcal{A}_\theta(x_i^-)) \tag{12}$$

where $\{x_i^-\}_{i=1}^{N_-}$ and $\{x_j^+\}_{j=1}^{N_+}$ are normal and anomalous samples, respectively, $\lfloor \cdot \rfloor$ indicates the flooring function, and $\mathcal{H}(\cdot)$ represents a hard-threshold function whose output is 1 for a positive argument and 0 otherwise.

### B.6.4 Implementation of Models

We utilize a standard autoencoder model provided by DCASE 2020 Challenge organizers and compare it to our approach. The model comprises a symmetrical arrangement of fully-connected layers in the encoder and decoder; 5 fully connected hidden layers in the input and 5 in the output, with 128-dimensional hidden layers and 32-dimensional latent space. In addition, we incorporate the IDNN model, which predicts the excluded frame using all frames except the central one instead of reconstructing the entire sequence. The IDNN model outperforms the autoencoder on non-stationary sound signals, i.e., sounds with short durations. The model consists of a encoder and decoder, which is similar to the components of an autoencoder but has an asymmetric layout; 3 fully-connected hidden layers that contract in dimension (64, 32, 16) are used in the encoder and 3 layers that expand in dimension (16, 32, 64) compose the decoder. The architectural design of each model follows the specifications outlined in their respective papers.

The MPDR models used for the experiment, MPDR-R and MPDR-IDNN, consists of an ensemble of autoencoders and an energy function, where the terms "R" and "IDNN" specify whether an autoencoder of IDNN is used to compute the energy, respectively. Each autoencoder consists of an encoder, spherical embedding layer, decoder; the encoder and decoder consist of 3 fully-connected layers each, with 1024-dimensional hidden space and a latent space with a dimension chosen among 32, 64, or 128. The autoencoder energy function used in the experiment is built using layers 5 fully-connected encoder layers and 5 decoder layers, whose hidden layers have 128 nodes and latent layer is composed of 32 nodes. The layers of IDNN are indentical to that of the autoencoder version, except for the input and output layers whose dimensions differ and add up to the total number of sampled frames. LMC hyperparameters used are listed in Table 7.

## C Empirical Guidelines for Implementing MPDR

Here, we present empirical tips and observations we found useful in achieving competitive OOD detection performance with MPDR. The list also includes heuristics that *did not* work when we tried.

- The training progress can be monitored by measuring an OOD detection metric (i.e., AUROC) computed between test (or validation) inliers and synthetic samples uniformly sampled over the autoencoder manifold $\mathcal{M}$. During a successful and stable run, this score tends to increase smoothly.
- Metropolis-style adjustment for LMC [64] did not improve OOD performance.
- The choice of activation function in the energy function affects the OOD detection performance significantly. We found that ReLU and LeakyReLU provide good results in general.
- In image datasets, stopping the training of the autoencoder manifold before convergence improves OOD detection performance.
- A longer Markov chains, both visible and latent, do not always lead to better OOD detection performance.

- A larger autoencoder does not always lead to better OOD detection performance.
- A larger energy function does not always lead to better OOD detection performance.

