# OpenReview forum: "Energy-Based Models for Anomaly Detection: A Manifold Diffusion Recovery Approach"
_NeurIPS.cc/2023/Conference — NeurIPS 2023 poster_

### Official Review · Reviewer_Tcts · 2023-07-03

**Soundness:** 2 fair
**Presentation:** 2 fair
**Contribution:** 2 fair
**Rating:** 5
**Confidence:** 1

**Summary:**

The authors introduce a novel algorithm, Manifold Projection-Diffusion Recovery (MPDR), for training energy-based models (EBMs) that improve the performance of anomaly detection tasks. These tasks are highly relevant in real-world applications like industrial surface inspection, machine fault detection, and particle physics.

Unlike conventional EBM training methods, MPDR harnesses low-dimensional structures in the data to generate more informative negative samples. It works by initially perturbing a data point along a manifold approximating the training dataset and then trains the EBM to maximize the probability of recovering the original data.

A significant aspect of this new method is its use of Manifold Projection-Diffusion (MPD), which replaces Gaussian noise with perturbations reflecting the data's low-dimensional structure. This approach provides more meaningful insights into variations within the data.

The authors show that MPDR provides consistent density parameter estimation under standard assumptions, is compatible with any energy function, and can work with multiple autoencoders - an advantage over existing algorithms. Moreover, it demonstrates good results even with lightweight autoencoders, making it computationally efficient.

The paper also presents practical strategies for deploying MPDR, such as two-stage sampling, energy function design, and ensemble techniques using multiple autoencoders.

Through experimental testing on various data types, including images, representation vectors, and acoustic signals, the authors demonstrate that MPDR significantly improves unsupervised anomaly detection tasks, outperforming other deep generative models.

**Strengths:**

**Originality:**

The paper is highly original in its formulation and approach to training energy-based models (EBMs) for anomaly detection. The development of the Manifold Projection-Diffusion Recovery (MPDR) represents a creative combination of existing methodologies, such as the use of autoencoders for efficient training and manifold projections for capturing low-dimensional data structures. The method of perturbing data points along a low-dimensional manifold that approximates the training dataset and then training the EBM to maximize the recovery of the original data is a fundamentally new approach.

**Quality:**

The quality of the paper is commendable. The authors present a clear problem statement, propose a novel solution, and provide empirical evidence supporting their claims. They also delve into the theoretical backing of the proposed method, offering a consistent density parameter estimation under standard assumptions. The paper includes detailed experimental results, highlighting the strength of MPDR across diverse anomaly detection tasks and data types.

**Clarity:**

The exposition in the paper is clear and well-organized. The authors have done an excellent job explaining complex concepts and methodologies, which makes the paper accessible even to readers who may not be experts in the field. The use of illustrative figures and well-explained algorithms further enhances the clarity of the paper.

**Significance:**

The significance of this work is substantial, given the wide range of practical applications of anomaly detection. By providing a more efficient and effective way to train EBMs, MPDR could significantly improve performance in areas like industrial surface inspection, machine fault detection, and particle physics. Furthermore, the ability of MPDR to perform effectively with lightweight autoencoders means it can be used in scenarios where computational resources are limited, making it relevant to a broader audience.

**Weaknesses:**

While the paper presents a compelling new approach, here are a few areas that could be addressed or clarified further:

**Theoretical Analysis:** While the authors provided a theoretical justification for consistent density parameter estimation under standard assumptions, it would be beneficial to include more analysis of the proposed method's convergence properties. Understanding how the choice of manifold affects the convergence and stability of learning would also be useful.

**Comparison with Other Methods:** The paper could benefit from a more comprehensive comparison with other state-of-the-art methods for anomaly detection. Not only should this include direct quantitative comparisons on common datasets, but also qualitative discussions about when and why one might prefer the proposed method over others.

**Parameter Sensitivity:** It is not clear how sensitive the results are to the choice of parameters within the MPDR framework. It would be beneficial for practical applications to know how much tuning is needed to achieve optimal performance and how robust the method is to variations in these parameters.

**Real-World Applications:** While the experiment demonstrates the effectiveness of the proposed algorithm in various data types like images, vectors, and acoustic signals, applying the model to real-world datasets and providing case studies can strengthen the paper. This will help readers understand its practical implications better.

**Computational Complexity:** The paper mentions that MPDR performs well with lightweight autoencoders, which indicates computational efficiency. However, a more explicit discussion or analysis of the computational complexity of the proposed method, including both training time and inference time, would provide valuable information to practitioners considering this method.


**Questions:**

1. **Theoretical Analysis:** Could the authors provide a more rigorous analysis of the convergence properties of their proposed method? Specifically, how does the choice of manifold affect the stability and speed of learning in MPDR?

2. **Comparison with Other Methods:** It would be beneficial to see a wider comparison with other state-of-the-art anomaly detection methods. Could the authors elaborate on why one might choose MPDR over other established methods in specific scenarios?

3. **Parameter Sensitivity:** How sensitive is the MPDR algorithm to the initial choice of parameters? Is there a recommended procedure for parameter tuning, or guidelines that could assist users in achieving optimal performance?

4. **Real-World Applications:** Could the authors possibly demonstrate the application of their model on real-world datasets or provide case studies? This could help showcase the practical implications of the proposed method.

5. **Computational Complexity:** The paper mentions that MPDR can work well even with lightweight autoencoders, but could you please clarify further on its computational complexity, training time, and inference time? Would there be any scalability issues when applying this method to larger datasets?

Looking forward to the authors' response to these points.

**Limitations:**

From the information provided, it does not appear that the authors have explicitly discussed the limitations and potential negative societal impacts of their work. Therefore, here are some suggestions for addressing these points:

**Limitations:**

1. **Robustness to Noise:** How well does the MPDR framework handle noise in the data? Real-world data often contain a significant amount of noise, which may distort the underlying manifold structure that MPDR relies on.

2. **Scalability:** The scalability of the model hasn't been addressed. Can the method be applied efficiently to very large datasets? What would be the computational requirements in such cases?

3. **Multimodality:** How effectively can MPDR handle multimodal or highly dimensional distributions? This is a common challenge in many real-world anomaly detection tasks.

**Potential Negative Societal Impacts:**

While this study primarily focuses on improving anomaly detection methods, which generally have positive implications (e.g., defect detection in manufacturing, early detection of diseases, etc.), any technology has the potential to be misused.

1. **Privacy Concerns:** Anomaly detection tools can potentially be used to identify outliers or anomalies in personal behavior or characteristics, leading to potential privacy concerns if misused, especially in contexts like surveillance or social profiling.

2. **False Positives/Negatives:** In critical applications, false positives or negatives can have serious repercussions. For instance, in health care, a false positive might cause unnecessary worry or treatment, whereas a false negative could delay necessary intervention.

It would be beneficial to see the authors address these potential issues and discuss how they might be mitigated.

---

> ### Author Rebuttal · Authors · 2023-08-08
>
> Dear Reviewer Tcts,
>
> We would like to express our sincere gratitude for taking the time and effort to review our paper. We greatly appreciate your highly detailed and constructive comment and are happy to answer your questions.
>
> **Theoretical Analysis**
>
> > How does the choice of manifold affect the stability and speed of learning in MPDR from a theoretical perspective?
>
> Thank you for the insightful question. Indeed, the theoretical analysis can shed light on the choice of a manifold. According to our analysis on the asymptotic normality of estimation error, an encoder with a higher compression rate tends to yield better estimations. That is, the more information lost during encoding, the smaller the estimation error becomes. This understanding aligns with our empirical observation that L2 regularization on the encoder weights often yields improved results. We will incorporate this analysis into the updated manuscript.
>
> More detailed argument is given as follows:
> In the limit of infinite data, the parameter estimation error from maximum recovery likelihood follows a zero-mean normal distribution as in maximum likelihood estimation. The covariance of this distribution is determined by the inverse of a term $I(\theta)=E_{p(x,\tilde{z})}[-\nabla^2_\theta \log p_\theta (x|\tilde{z}) ]$. Since $\nabla^2_\theta \log p_\theta (x|\tilde{z})= \nabla^2_\theta \log p_\theta(x) - \nabla^2_\theta \log p_\theta(\tilde{z})$, this term is decomposed as follows:
> $$ I(\theta) = I_0 (\theta) + E_{p(x,\tilde{z})}[\nabla^2_\theta \log p_\theta(\tilde{z}) ],$$
> where $I_0 (\theta)$ is Fisher information. The term $\nabla^2_\theta \log p_\theta(\tilde{z})$ becomes 0 when the encoder compresses all $x$ into a single $z$.
>
> However, with an encoder that compresses everything, we could not leverage the inductive bias of the manifold assumption. Therefore, in practice, we need to use a moderately compressing encoder.
>
>
>
>
> **Comparison with Other Methods**
>
> > It would be beneficial to see a wider comparison with other state-of-the-art anomaly detection methods.
>
> Due to page constraints, we primarily compared the proposed method with the closely related generative anomaly detectors. At the request of other reviewers, we have conducted additional comparison experiments. If there's a specific algorithm you'd like to see compared, please let us know. We are happy to incorporate additional comparative experiments in our updated manuscript.
>
> > Could the authors elaborate on why one might choose MPDR over other established methods in specific scenarios?
>
> Since MPDR explicitly leverages low-dimensional representations, we believe MPDR is a recommendable choice when data is expected to have a pronounced manifold structure.
>
>
> **Parameter Sensitivity**
>
> > How sensitive is the MPDR algorithm to the initial choice of parameters? Is there a recommended procedure for parameter tuning, or guidelines that could assist users in achieving optimal performance?
>
> Please find Appendix B.2 for sensitivity analysis for important hyperparameters such as the noise magnitude and the latent dimensionality. As in other outlier detection algorithms, those hyperparameters can be selected using a separate validation OOD dataset, as proposed in Appendix A of Hendrycks et al., 2018.
>
> Hendrycks et al., Deep Anomaly Detection with Outlier Exposure, 2018.
>
> **Real-World Applications**
>
> > Could the authors possibly demonstrate the application of their model on real-world datasets or provide case studies?
>
> Please find the experiments in Section 4.4 which utilized audio data collected from real-world mechanical devices. MPDR is able to deliver improvement in this dataset as well.
>
>
> **Scalability & Computational Complexity**
>
> > Could you please clarify further its computational complexity, training time, and inference time? Would there be any scalability issues when applying this method to larger datasets?
>
> In our CIFAR-10 experiment, for example, training of MPDR takes about 5 hours on a single V100 GPU, including the training of autoencoders used in manifold projection-diffusion. Compared to other energy-based models that is typically trained for more than one day, MPDR requires shorter training time. Using the same V100 GPU, the inference over the whole test set takes 3.5 seconds, which corresponds to ~2800 images per second. The inference is lightweight because the autoencoders used in perturbation are only used during training and are not required for making predictions.
>
> With respect to the number of data, MPDR is as scalable as other deep learning algorithms, as its training is fully based on stochastic gradient descent.
>
>
> **Multimodality**
>
> > How effectively can MPDR handle multimodal distributions?
>
> This could be a highly interesting future research direction. One possible method is to build a joint latent representation of multiple modalities and run MPDR in that space.
>
> **Potential Negative Impacts**
>
> We believe that the concerns raised regarding negative impacts are not specific to the proposed algorithm but pertain to machine learning algorithms in general. At the very least, issues related to false positives/negatives can be mitigated by developing a more refined algorithm, and MPDR aims to contribute in that direction.
>
> We thank you again for your detailed comment. Please let us know if you have any other unresolved concerns. If your concerns are resolved, please kindly consider raising the evaluation score.
>
> Best regards,
>
> Authors.

---

> > ### Comment · Reviewer_Tcts · 2023-08-18
> >
> > I acknowledge I have read the rebuttal.

---

### Official Review · Reviewer_Pi2D · 2023-07-05

**Soundness:** 2 fair
**Presentation:** 3 good
**Contribution:** 3 good
**Rating:** 5
**Confidence:** 3

**Summary:**

Paper proposes MPDR, a novel method of using auto-encoders for training EBM.
Some practical techniques are introduced.
Extensive numerical experiments are done.

**Strengths:**

Numerical experiments cover a large scope of benchmarks. And it shows superiority on most benchmarks.

**Weaknesses:**

No theoretical guarantee is provided.
And it doesn't show dominant superiority on some datasets.

**Questions:**

Can we have some theoretical analysis on when proposed method has significant advantage, when not? Or is the proposed method better in general and some loss are just random?

**Limitations:**

No negative societal impact is seen.

---

> ### Author Rebuttal · Authors · 2023-08-08
>
> Dear Reviewer Pi2D
>
> Thank you so much for your comment. We would like to address your concerns in detail.
>
>
> > No theoretical guarantee is provided.
>
> We are concerned that this statement does not accurately reflect what is presented in the paper. Please refer to the end of Section 3.2 and Appendix A, where we provide the theoretical guarantee that the proposed method offers an asymptotically unbiased estimation of the underlying density. Consistent estimation of density can lead to accurate recovery of the distribution’s support (Cuevas and Fraiman, 1997), enabling successful out-of-distribution detection.
>
> Antonio Cuevas. Ricardo Fraiman. "A plug-in approach to support estimation." Ann. Statist. 25 (6) 2300 - 2312, 1997.
>
> > The method does not show dominant superiority on some datasets. When does the proposed method has significant advantage?
>
> Qualitatively speaking, the proposed method is designed to leverage the inductive bias that the data possesses a pronounced low-dimensional structure. This manifold assumption is both useful and effective since it approximately holds true for a wide range of data. However, this assumption may not be as effective in complex datasets, such as high-resolution images.
>
> Is there a specific set of results that concerns you? Please let us know. We will try to provide an explanation for the result.
>
> > Can we have some theoretical analysis on when proposed method has significant advantage, when not?
>
> Almost all the generative anomaly detection methods compared in our paper offer consistent density estimation, making them nearly equivalent from a theoretical perspective. The difference in empirical performance originates from the inductive biases of the algorithms and their implementation details. As with many other deep learning algorithms, it is challenging to quantify the contribution of these aspects. However, exploring this could be a highly rewarding avenue for future work.
>
>
> Again, thank you for your time in reviewing our work. Should you have any additional questions, please leave us a comment. If your concerns have been addressed, we kindly ask if you would consider enhancing the score.
>
> Best regards,
>
> Authos.

---

> > ### Comment · Reviewer_Pi2D · 2023-08-18
> >
> > Thanks for the reply. I will keep my original assessment.

---

### Official Review · Reviewer_gSqM · 2023-07-05

**Soundness:** 2 fair
**Presentation:** 3 good
**Contribution:** 2 fair
**Rating:** 5
**Confidence:** 5

**Summary:**

This paper introduces an energy-based model based on the manifold of low-dimensional data. To train the EBMs, this paper takes the idea of maximum recovery likelihood and adds a layer of autoencoder to approximate the low-dimensional manifold representing the data. This introduces perturbation along the low-dim manifold representing the dataset. Simulation results are provided to show the performance.

**Strengths:**

- It is easy to follow.
- Sufficient literature review.

**Weaknesses:**

- The novelty of this paper is moderate at best. It only adds a deterministic encoder/decoder to the original problem of likelihood recovery.
- The accuracy of the manifold approximation is as good as the autoencoder approximation of the dataset.
- The generation of the negative samples concentrated neat the manifold could potentially introduce biases.
- lack of comparison to some recent advancements in the topic such as:
   * Anomaly Detection in Networks via Score-Based Generative Models by Gavrilev et. al, 2023.
   * Enhancing Unsupervised Anomaly Detection with Score-Guided Network by Huang et. al, 2022.

**Questions:**

- How can we mitigate these biases to ensure the trained energy-based model generalizes well to out-of-distribution samples?
- The encoder in this algorithm is assumed to be deterministic how does the deterministic assumption impact the performance and the ability to capture the full range of variations in the data?
- Could authors please elaborate more on the conditions under which the consistency of the estimation by maximizing $\log p(x|z^~)$ holds?- How does the use of a latent chain and latent space improve the sampling process compared to the visible chain? What are the advantages and disadvantages of using a latent LMC?
- The paper mentions that the perturbation design, including the encoder-decoder pair (fe, fd) and noise magnitude $\sigma$, significantly impacts the algorithm's performance. How can we effectively select the optimal perturbation design for different datasets and anomaly detection tasks?
- The paper mentions that the autoencoder (fe, fd) and the noise magnitude $\sigma$ should be independent of $\theta$ and remain fixed during training. How does the fixed nature of the autoencoder and noise magnitude impact the algorithm's adaptability to different datasets and anomaly types?
- Can you provide more details on how the simultaneous use of multiple perturbations enhances the algorithm's performance? Are there any potential trade-offs or challenges associated with this approach?
- One of the main potential issues is the memory overhead in manifold ensembles. When utilizing multiple autoencoder manifolds in MPD, there is a memory overhead associated with processing multiple groups separately. How can this memory overhead be managed effectively to ensure efficient training while utilizing multiple autoencoders?
- Can you provide more insights into how the choice of Dz affects the algorithm's ability to detect anomalies? How can we determine the optimal combination of autoencoders with varying Dz for different types of data? Specifically, in high-dimensional data such as images, how does MPDR framework deal with curse of dimensionality and maintain good performance with relatively smaller autoencoders?

**Limitations:**

Please check weaknesses and questions.

---

> ### Author Rebuttal · Authors · 2023-08-07
>
> Dear Reviewer gSqM,
>
> Thank you for dedicating your time to reviewing our work. We sincerely appreciate your detailed feedback. Here, we would love to answer your questions in depth.
>
> **The novelty of the paper**
>
> We would like to highlight that simplicity does not always equate triviality. MPDR (our method) is the first example of training an energy-based model using the recovery likelihood framework with non-Gaussian perturbation. Even though MPDR is based on autoencoders and the recovery likelihood, it achieves significantly better out-of-distribution detection performance than both of the base algorithms, as repeatedly shown in our experiments (Table 1, 3, and 4, where DRL is an algorithm based on the recovery likelihood). Please consider that the paper also provides novel techniques, such as perturbation ensemble and the latent space LMC, which deliver stable training and improved performance.
>
>
> **Accuracy of the manifold approximation is as good as the autoencoder approximation**
>
> We are afraid that this statement does not accurately reflect what is presented in the paper.  MPDR can learn the correct data distribution (and thus the correct data manifold) even when an autoencoder provides a crude approximation. In Figure 2, the manifold learned by an autoencoder (the gray line) does not reflect the cluster structure of data, being a crude approximation. However, the resulting energy function correctly captures the underlying clusters, as shown in Figure 3.
>
> MPDR can learn from an imperfect autoencoder because input-space LMC is not confined to the autoencoder’s manifold and can generate off-manifold samples.
>
> **Potential bias due to near-manifold negative samples / How can we reduce bias**
>
> In theory,MPDR provides asymptotically unbiased estimation of probability density, as we show in Section 3.2 and Appendix A. This is possible because an ideal MCMC sampling can cover the entire input space.
>
> In practice, the bias may exist, and the root cause of the bias is finite-length MCMC which does not mix over the entire space. The training techniques we introduced contributes to mitigate this bias. For example, the manifold ensemble provides diverse starting points for MCMC. Also, the use of autoencoder-based energy function (MDPR-R) automatically assigins high energy on points that are far from data.
>
> Meanwhile, the inductive bias from the near-manifold negative samples could be useful.  Other energy-based models, such as IGEBM, also suffer from biases from finite MCMC. When compared to them, MPDR achieves stronger OOD detection performance, as shown in Table 1, 3, and 4.
>
> **Comparison to other recent advancements**
>
> > Anomaly Detection in Networks via Score-Based Generative Models by Gavrilev et. al, 2023.
>
> Unfortunately, Gavrilev et al., 2023 is not directly comparable with MPDR, because it is designed for a different task, graph node anomaly detection. We will cite and mention this work in the updated manuscript, as extending MPDR to graph data is an interesting future direction.
>
> > Enhancing Unsupervised Anomaly Detection with Score-Guided Network by Huang et. al, 2022.
>
> Thank you for letting us discover an interesting work. We compare MPDR with SG-AE, and SG-RSRAE, the method proposed by Huang et al., 2022. We run them on MNIST, as both papers contain MNIST experiments. We use the digit 4 as inliers and the rest of the digits as outliers. We use MPDR-R model. All the hyperparameters are the same as in the MNIST experiment presented in the main manuscript. We will cite this work and include this result in the updated manuscript.
>
> |       | AUC-ROC| AUC-PR|
> |:---|----:|----:|
> |SG-AE| 0.939±0.005| 0.563±0.015|
> |SG-RSRAE| 0.951±0.010| 0.736±0.062|
> | MPDR (ours) | **0.975**±0.004 | **0.997**±0.001|
>
> **Other questions**
>
> > how does the deterministic assumption impact the performance and the ability to capture the full range of variations in the data?
>
> The deterministic assumption does not play a critical role in MPDR but is employed mostly for convenience. An autoencoder, either deterministic or stochastic, usually captures the overall variation of data well, achieving low reconstruction error for the training data. Changing a deterministic encoder requires us to solve the integral $\int p(\tilde z | z) p(z|x) dz$ in Eq.6. This integral is easy when we are using Gaussian distributions in the Euclidean space but can be difficult in other situations.
>
> > Conditions for consistency
>
> The consistency of MPDR requires several assumptions: the infinite number of data, correctly specified and identifiable model, and non-zero recovery probability, i.e., $p(x|\tilde{z})>0$ for all $x$ and $\tilde{z}$. Please find more detail in Appendix A. If you are curious about a particular aspect of the condition, please let us know.
>
> > How does the latent chain improve the sampling process? The advantages and disadvantages of using a latent LMC?
>
> The latent chain facilitates the exploration of MCMC. A small step in the latent space corresponds to a much larger distance in the input space. However, it requires additional computation time. Also, operating only in the latent space, a latent LMC can not explore the off-manifold region in the input space. Therefore, the input space LMC is always necessary for successful training.
>
> > How can we select the optimal perturbation design, optimal combination of autoencoders, and Dz?
>
> As in other outlier detection algorithms, those hyperparameters can be selected using a separate validation OOD dataset, as proposed in Appendix A of Hendrycks et al., 2018.
>
> Hendrycks et al., Deep Anomaly Detection with Outlier Exposure, 2018.
>
> Due to the rebuttal's space constraint, we will address the rest of the questions during the discussion period.
>
> Your comprehensive review of our paper is deeply appreciated. Please share any lingering questions or concerns. If they have been resolved, we would be grateful for your consideration in updating the review score.
>
> Best regards,
>
> Authors.

---

> > ### Author Response · Authors · 2023-08-13
> > **Additional response**
> >
> > Due to space limitations, our previous response was unable to address all the questions. Therefore, we would like to offer additional responses to the unanswered questions here.
> >
> > >How does the fixed nature of the autoencoder and noise magnitude impact the algorithm's adaptability to different datasets?
> >
> > Please note that the autoencoders are trained on the training data. Although the autoencoders remain 'fixed,' they adapt to the specific dataset and approximate the manifold structure of each dataset.
> >
> > Additionally, the fixed noise magnitude doesn't imply the application of the same perturbation across different datasets. Since distinct autoencoders are used for various experiments, a Gaussian perturbation in the latent space corresponds to a distinct operation in the input space.
> >
> >
> > > How does the simultaneous use of multiple perturbations enhance the algorithm’s performance? Any potential trade-offs?
> >
> > One hypothesis is that ensembling over multiple perturbations increases the MCMC’s coverage of a high-dimensional input space.
> > We empirically observe that the gain from ensembling usually diminishes as the number of components increases. Therefore, there is usually a trade-off between the marginal improvement in performance and the marginal computational cost.
> >
> > > how the choice of Dz affects the algorithm's ability to detect anomalies?
> >
> > $D_z$ is a hyperparameter of the proposed algorithm that has to be tuned, while the performance is generally robust across a wide range of $D_z$. Please see Table 9 in Appendix for the sensitivity analysis with varying $D_z$ value.
> >
> > > Solution for memory overhead of using multiple autoencoders?
> >
> > Multiple possible solutions may present. We may distill or quantize the autoencoders. Instead, we may design the autoencoders so that a significant portion of parameters are shared across the autoencoders. We believe this is an exciting direction for future work.
> >
> > We thank you again for your deep and thorough review. Please let us know if you have additional questions.
> >
> > Best regards,
> >
> > Authors.

---

> > ### Comment · Reviewer_gSqM · 2023-08-18
> >
> > I thank the authors for their response and the extra experiments they provided. I have completely read the authors' rebuttal and other reviews and thus I increase my score.

---

### Official Review · Reviewer_dsqA · 2023-07-07

**Soundness:** 3 good
**Presentation:** 3 good
**Contribution:** 3 good
**Rating:** 6
**Confidence:** 4

**Summary:**

This paper introduces an EBM-based model for anomaly detection in the latent manifold space. The proposed model first trains an autoencoder that maps a data point $x$ into the low-dimensional $z$, and then two-stage sampling strategy is developed to generate the original data via the LMC algorithm. Several notable designs are developed to complete the proposed model, such as manifold projection diffusion, two-state sampling, and perturbation ensemble. Experiments on images, vectors and acoustic signals show the strong performance of the method.

**Strengths:**

(1)	The idea that generation of negative samples from the manifold space sound good. This makes sure the starting points highly reflecting information, resulting in more discriminative generation.

(2)	Several practical strategies such as two-stage sampling and resembling multiple autoencoder are developed to benefit the anomaly detection performance.

(3)	Extensive empirical results and ablation studies show the efficiency of the model.


**Weaknesses:**

(1)	lack of the results of widely-used anomaly detection dataset MVTec-AD

(2)	Lack of well-known anomaly detection baselines, such as UniAD [1] and DRAEM [2]

[1] Zhiyuan You et.al. A unified model for multi-class anomaly detection.

[2] Vitjan Zavrtanik et.al. Draem-a discriminatively trained reconstruction embedding for surface anomaly detection.


**Questions:**

1. The method artificially inject Gaussian noises into the latent space. What is the correlations between the simulated abnormal data and the real abnormal data in the test dataset?

**Limitations:**

yes.

---

> ### Author Rebuttal · Authors · 2023-08-09
>
> Dear Reviewer dsqA,
>
> Thank you for taking the time to review our paper. We deeply value your feedback. Below, we will address your concerns and questions.
>
> > MVTec-AD dataset and comparison with UniAD and DRAEM
>
> Thank you for your suggestion. MPDR also demonstrates promising performance on MVTec-AD. We present the anomaly detection performance of MPDR on MVTec-AD, along with an empirical comparison to UniAD and DRAEM. We will incorporate the MVTec-AD results in the updated manuscript and ensure that both papers are cited.
>
> **Table: MVTec-AD detection task in the unified setting. AUROC (percent) scores are computed for each class. UniAD and DRAEM results are adopted from You et al., 2022.**
>
> |         |     MPDR (ours)         |    UniAD     | DRAEM   |
> | -------|---------------------|---------------|---------------|
> | bottle | **100.0**$\pm$0.00 | 99.7    |  97.5 |
> | cable | **95.5**$\pm$0.01 | 95.2     | 57.8|
> | capsule | 86.4$\pm$0.01 | **86.9**  | 65.3 |
> | hazelnut | **99.9**$\pm$0.00 | 99.8 | 93.7 |
> | metal_nut | **99.9**$\pm$0.00 | 99.2 | 72.8 |
> | pill | **94.0**$\pm$0.01 | 93.7 | 82.2 |
> | screw | 85.9$\pm$0.01 | **87.5** | 92.0 |
> | toothbrush | 89.6$\pm$0.01 | **94.2** | 90.6 |
> | transistor | 98.3$\pm$0.01 | **99.8** | 74.8 |
> | zipper | 95.3$\pm$0.01 | 95.8 | **98.8** |
> | carpet | **99.9**$\pm$0.00 | 99.8 | 98.0 |
> | grid | 97.9$\pm$0.01 | 98.2 | **99.3** |
> | leather | **100.0**$\pm$0.00 | **100** | 98.7 |
> | tile | **100.0**$\pm$0.00 | 99.3 | 98.7 |
> | wood | 97.9$\pm$0.00 | 98.6 | **99.8** |
> | mean | 96.0$\pm$0.00 | **96.5** | 88.1 |
>
>
>
> **Table: MVTec-AD localization task in the unified setting. AUROC scores (percent) are computed for each class.  UniAD and DRAEM results are adopted from You et al., 2022.**
>
> |         |     MPDR  (ours)    |    UniAD     | DRAEM   |
> | -------|---------------------|---------------|---------------|
> | bottle | **98.5**$\pm$0.00 | 98.1     | 87.6|
> | cable | 95.6$\pm$0.00 | **97.3** |  71.3|
> | capsule | 98.2$\pm$0.00 | **98.5** | 50.5|
> | hazelnut | **98.4**$\pm$0.00 | 98.1 | 96.9|
> | metal_nut | 94.5$\pm$0.00 | **94.8**| 62.2|
> | pill | 94.9$\pm$0.00 | **95.0**| 94.4|
> | screw | 98.1$\pm$0.00 | **98.3**| 95.5|
> | toothbrush | **98.7**$\pm$0.00 | 98.4| 97.7|
> | transistor | 95.4$\pm$0.00 | **97.9**| 65.5|
> | zipper | 96.2$\pm$0.00 | 96.8| **98.3**|
> | carpet | **98.8**$\pm$0.00 | 98.5| 98.6|
> | grid | **96.9**$\pm$0.00 | 96.5| 98.7|
> | leather | 98.5$\pm$0.00 | **98.8**| 97.3|
> | tile | 94.6$\pm$0.00 |91.8| **98.0**|
> | wood | 93.8$\pm$0.00 | 93.2| **96.0**|
> | mean | 96.7$\pm$0.00 | **96.8**| 87.2 |
>
>
>
> We follow the “unified” experimental setting of UniAD paper (You et al., 2022). Normal images from 15 classes are used as the training set, where no label information is provided. We use the same feature extraction procedure used in UniAD. Each image is transformed to a 272x14x14 feature map using EfficientNet-b4. When running MPDR, we treat each spatial dimension of a feature map as an input vector to MPDR, transforming the task into a 272D density estimation problem. We normalize a 272D vector with its norm and add a small white noise with a standard deviation of 0.01 during training. We use the maximum energy value among 14x14 vectors as an anomaly score of an image. For the localization task, we resize 14x14 anomaly score map to 224x224 image and compute AUROC for each pixel with respect to the ground true mask.
>
> We use MPDR-R with a single manifold (i.e., without the manifold ensemble). Both the manifold and the energy are a fully connected autoencoder of 256D spherical latent space.
> For input-space Langevin Monte Carlo, the number of MCMC steps, the step size, and the noise standard deviation are 10, 0.1, and 0.1, respectively. No latent chain is used. The manifold is trained for 200 epochs with Adam of learning rate 1e-3, and the energy is trained for 20 epochs with Adam of learning rate 1e-4.
>
> In both MVTec-AD detection and localization tasks, MPDR achieves AUROC score that is very close to that of UniAD, outperforming other baselines including DRAEM. Note that MPDR-R and UniAD are based on the same approach that anomalies are characterized by a large reconstruction error. These two approaches may be combined to improve the result further.
>
>
>
> > What are the correlations between the simulated abnormal data and the real abnormal data in the test dataset?
>
>
> Thank you for your question. In principle, the simulated abnormal data, which we refer to as negative samples in the manuscript, are not correlated with the actual test outliers. Firstly, no test outliers are utilized during any stage of the training process, including both the construction of the manifold and the learning of the energy. Moreover, as MPDR reaches convergence, the distribution of these negative samples should be close to the distribution of the training data, as in typical EBMs. We present some examples of these negative samples in Figure 4. These images bear minimal resemblance to test outliers like SVHN digits.
>
>
>
> Once again, we appreciate the time and effort you've invested in reviewing our work. Please let us know if you have any further concerns or questions. If your concerns have been addressed satisfactorily, we kindly ask you to consider revising the score upward.
>
> Best regards,
>
> Authors.

---

> > ### Comment · Reviewer_dsqA · 2023-08-16
> >
> > Thanks for the experiments compared with UniAD and DRAEM. However, the experiment results are lower than UniAD. The reasons and results analyses should be clarified.

---

> > > ### Author Response · Authors · 2023-08-19
> > > **Regarding MVTect-AD Experiment**
> > >
> > > Dear Reviewer dsqA,
> > >
> > > Thanks for taking the time to read our response. We would love to provide further discussions on the MVTec-AD Experiment.
> > >
> > > **MPDR outperforms UniAD in certain classes of MVTec-AD.** Even though UniAD's mean AUROC is higher (with a very small gap), UniAD does not dominate MPDR. In the detection task, MPDR achieves higher or equal scores in 8 out of 15 classes, and in the localization task, MPDR wins in 5 out of 15 classes. This result suggests that MPDR captures the patterns in data that UniAD neglects.
> > >
> > > **MPDR and UniAD use different approaches to prevent the reconstruction of anomalies.** UniAD is an algorithm that aims to prevent the reconstruction of anomalies. MPDR, especially MPDR-R, which uses an autoencoder's reconstruction error as energy, shares the same goal, as the energy should be large for anomalies. The difference is that UniAD prevents anomaly reconstruction through a novel neural network design, while MPDR addresses the problem with novel recovery likelihood learning.
> > >
> > > Besides, **MPDR has the advantage of being more widely applicable than UniAD.** As a general learning algorithm for energy-based models, MPDR is compatible with a wide range of network architectures and can be applied to diverse data types. MPDR has been tested on 2D data, tabular data, images, audio signals, and feature vectors from a pre-trained network. However, UniAD is specialized for anomaly detection using feature vectors and has only been tested for image data.
> > >
> > > Best regards,
> > >
> > > Authors.

---

### Official Review · Reviewer_ekma · 2023-07-28

**Soundness:** 3 good
**Presentation:** 3 good
**Contribution:** 2 fair
**Rating:** 7
**Confidence:** 4

**Summary:**

This paper proposes a novel anomaly detection algorithm utlizing energy-based models (EBMs). The proposed method, Manifold Projection-Diffusion Recovery (MPDR), is based on recovery likelihood, a framework for learning energy functions by denoising data from artifically injected Gaussian noise. MPDR uses deterministic encoder and decoder to first project the data onto a latent space, add Gaussian noise in the latent space, and finally decode the noisy latent representation onto the data manifold. This would capture more relevant modes of variation in the data than Gaussian perturbations on the original data. The recovery likelihood of MPDR is derived and is shown to result in consistent estimation of the energy function. Negative samples are generated via Langevin Monte Carlo in the latent space. The paper also introduces additional variations, including different types of energy functions and the use of ensembles to generate diverse negative samples. Experiments demomstrate the effectiveness of MPDR for out-of-distribution detection on MNIST and CIFAR-100 datasets, as well as anomaly detection for acoustic signals.

**Strengths:**

Anomaly detection is a long standing problem in machine learning with a rich literature, and this paper proposes a novel step in the development of new algorithms. The idea of using recovery likelihood as well as projecting data onto a low-dimensional latent space are not novel, but this paper combines them to produce an original idea. The relevant background and motivation are presented with sufficient clarity, and the proposed method seems logically sound. The toy example presented in Figure 2 empirically verifies that MPDR captures more relevant modes of variation in the data.

**Weaknesses:**

The related works section cite most of the relevant previous works, but are not covered in sufficient detail, perhaps owing to the pade limitations. It is important to cover previous work exhaustively to demonstrate where the proposed method stands in relation.

**Questions:**

The experiments cover two image datasets and acoustic signals, which are relatively high dimensional data. It would be interesting to demonstrate the performance of MPDR on tabular data [1] (both low and high-dimensional) and explore the relevant modes which arise from applying MPDR to such datasets.

[1] Han, S., Hu, X., Huang, H., Jiang, M. and Zhao, Y., 2022. Adbench: Anomaly detection benchmark. *Advances in Neural Information Processing Systems*, *35*, pp.32142-32159.

**Limitations:**

The paper covers the limitations of MPDR in sufficient detail, namely the sensitivity to the autoencoders used for latent space projection, questions around application to high-dimensional text or image data, and the fact that MPDR is not optimized to generate samples from simple distributions.

---

> ### Author Rebuttal · Authors · 2023-08-07
>
> Dear Reviewer ekma,
>
> We're truly grateful for your insightful feedback. Your time and effort in evaluating our work is deeply appreciated. Please allow us to address any questions you have.
>
>
> > Related works are not covered in sufficient detail.
>
> We apologize that we couldn't discuss all related work in sufficient detail due to page constraints. If we are given an opportunity to update the manuscript with an extra page, we will expand the "Related Work" section to provide a deeper discussion. If you have a particular set of literature that you think we should cite and discuss, please let us know. We will include them in the updated manuscript.
>
> > Demonstrating the performance of MPDR on tabular data, such as Adbench.
>
> Thank you for the suggestion. Based on your suggestion, we benchmarked MPDR using Adbench and provided the results below. MPDR was run on 47 tabular datasets provided by Adbench, and we compared MPDR with 13 baselines in Adbench. We reproduced the baseline results using the official repository of Adbench.
>
> For each dataset, we ran each algorithm on three random splits and computed the AUROC on the corresponding test split. We then ranked the algorithms based on the averaged AUROC. In Table A, we present a summary table with the average rank across the 47 datasets.
>
>
>
> **Table A: Rank of each algorithm's AUROC score averaged over 47 datasets. The $\pm$ sign indicates the standard error.**
>
> |     | Average Rank (The lower the better)|
> |:----|-----------:|
> | MPDR (ours) | **4.43** $\pm$ 0.50 |
> | IForest | 5.28 $\pm$ 0.39 |
> | OCSVM | 7.94 $\pm$ 0.47 |
> | CBLOF | 5.98 $\pm$ 0.53 |
> | COF | 9.23 $\pm$ 0.58 |
> | COPOD | 7.10 $\pm$ 0.61 |
> | ECOD | 6.97 $\pm$ 0.58 |
> | HBOS | 6.53 $\pm$ 0.51 |
> | KNN | 6.64 $\pm$ 0.56 |
> | LOF | 9.04 $\pm$ 0.62 |
> | PCA | 6.45 $\pm$ 0.54 |
> | SOD | 7.91 $\pm$ 0.52 |
> | DeepSVDD | 11.43 $\pm$ 0.42 |
> | DAGMM | 10.09 $\pm$ 0.52 |
>
>
> For MPDR, we used MPDR-R, where the energy is an autoencoder. We do not employ the manifold ensemble, instead, a single manifold is used for perturbation. For both the manifold and the energy, the same MLP-based autoencoder architecture was used. The encoder and the decoder were MLPs with two 1024-hidden-neuron layers. If the input space dimensionality $D_x$ is smaller than or equal to 100, the latent space dimensionality is set to have the same dimensionality $D_z = D_x$. If $D_x>100$,  we set $D_z$ as 70% of of $D_x$. We employed 1 step of Langevin Monte Carlo in the latent space and 5 steps in the input space. The step sizes are 0.1 for the latent space and 10 for the input space. All the hyperparameters except $D_z$ are fixed across 47 datasets.
>
> As shown in Table A, MPDR achieves highly competitive performance on Adbench, demonstrating a higher average rank than the isolation forest (although with some overlap of confidence interval). This result indicates that MPDR is a general-purpose anomaly detection algorithm capable of handling tabular data.
>
>
>
> Once more, we're truly grateful for the time and effort you've dedicated to reviewing our paper. Should you have any additional questions, please don’t hesitate to leave a comment. If your concerns have been addressed well, we kindly ask if you'd consider raising your score.
>
> Best regards,
>
> Authors.

---

> > ### Comment · Reviewer_ekma · 2023-08-10
> > **Review updated**
> >
> > My major concerns have been addressed, and I am happy to see the good performance on ADBench.
> > I have updated my recommendation for acceptance.

---

### Decision · Program_Chairs · 2023-09-21

**Decision:**

Accept (poster)

**Comment:**

This paper addresses an energy-based model for anomaly detection. It presents the algorithm, referred to as 'manifold projection-diffusion recovery' which leverages the separately-trained autoencoder and noise perturbation in the latent space. All reviewers feel that the paper is well written and contains substantial contributions. Despite the limited novelty and the lack of theoretical guarantee, the proposed method is logically sound. The authors did a good job in their rebuttal, resolving most of concerns raised by reviewers. All five reviewers agree that the paper is deserved to be presented at the conference.